# BoltzNCE: Learning Likelihoods for Boltzmann Generation with Stochastic Interpolants and Noise Contrastive Estimation

**Rishal Aggarwal**
CMU-Pitt Computational Biology
Dept. of Computational & Systems Biology
University of Pittsburgh
Pittsburgh, PA 15260
rishal.aggarwal@pitt.edu

**Jacky Chen**
CMU-Pitt Computational Biology
Dept. of Computational & Systems Biology
University of Pittsburgh
Pittsburgh, PA 15260
jackychen@pitt.edu

**Nicholas M. Boffi**
Machine Learning Dept.
Dept. of Mathematical Sciences
Carnegie Mellon University
Pittsburgh, PA 15213
nboffi@andrew.cmu.edu

**David Ryan Koes**
Dept. of Computational & Systems Biology
University of Pittsburgh
Pittsburgh, PA 15260
dkoes@pitt.edu

## Abstract

Efficient sampling from the Boltzmann distribution given its energy function is a key challenge for modeling complex physical systems such as molecules. Boltzmann Generators address this problem by leveraging continuous normalizing flows to transform a simple prior into a distribution that can be reweighted to match the target using sample likelihoods. Despite the elegance of this approach, obtaining these likelihoods requires computing costly Jacobians during integration, which is impractical for large molecular systems. To overcome this difficulty, we train an energy-based model (EBM) to approximate likelihoods using both noise contrastive estimation (NCE) and score matching, which we show outperforms the use of either objective in isolation. On 2D synthetic systems where failure can be easily visualized, NCE improves mode weighting relative to score matching alone. On alanine dipeptide, our method yields free energy profiles and energy distributions that closely match those obtained using exact likelihoods while achieving $100\times$ faster inference. By training on multiple dipeptide systems, we show that our approach also exhibits effective transfer learning, generalizing to new systems at inference time and achieving at least a $6\times$ speedup over standard MD with only a bit of fine-tuning. While many recent efforts in generative modeling have prioritized models with fast *sampling*, our work demonstrates the design of models with accelerated *likelihoods*, enabling the application of reweighting schemes that ensure unbiased Boltzmann statistics at scale. Our code is available at https://github.com/RishalAggarwal/BoltzNCE.

## 1 Introduction

Obtaining the equilibrium distribution of molecular conformations defined by an energy function is a fundamental problem in the physical sciences [1–3]. The Boltzmann distribution describes the equilibrium probability density and is given by $p(x) \propto \exp(-E(x)/K_B T)$ where $E(x)$ is the energy of molecular conformer $x$, $K_B$ is the Boltzmann constant, and $T$ is temperature. Sampling from this distribution is particularly difficult for molecular systems due to the non-convex nature of the energy

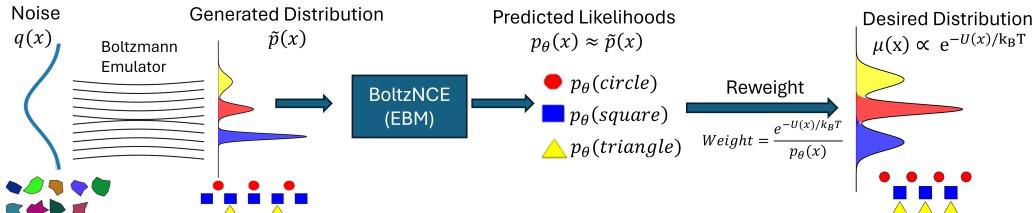

Figure 1: **Overview.** BoltzNCE offers an accelerated alternative to exact model likelihood computation. Samples from a prior are first transformed to a distribution of conformers by a Boltzmann Emulator, which is easy to sample from but difficult to evaluate likelihoods for. The generated samples are then reweighted with likelihoods estimated by an energy-based model (EBM), which we train to approximate the emulator distribution with a hybrid score matching and noise contrastive estimation scheme. This EBM gives access to likelihoods in a single function call, enabling us to reweight to the target Boltzmann distribtion up to $100\times$ faster than exact computation.

landscape, leading to the presence of widespread energy basins, metastability, and slow transitions. Traditional approaches for sampling conformers, such as Markov chain Monte Carlo (MCMC) and molecular dynamics (MD) [4, 5], often get trapped in these energy wells, which necessitates long simulation timescales to produce uncorrelated samples [6]. Consequently, it is particularly inefficient to obtain samples from independent metastable states.

In recent years, several generative deep learning methods have been proposed to address the molecular sampling problem. One prominent class is Boltzmann Generators (BGs) [7–10], which transform a simple prior distribution (e.g., a multivariate Gaussian) into a distribution over molecular conformers that can be reweighted to approximate the Boltzmann distribution. When reweighting is not applied, the model is referred to as a Boltzmann Emulator [8], whose primary aim is to efficiently sample metastable states of the molecular ensemble. While often qualitatively reasonable, Boltzmann Emulators alone fail to recover the true Boltzmann distribution and require the reweighting step for exact recovery of the system's equilibrium statistics.

To compute the likelihoods of generated samples, BGs are constrained to the class of normalizing flows. While earlier methods built these flows using invertible neural networks [7, 11], more recent approaches prefer using continuous normalizing flows (CNFs) [8, 12] due to their enhanced expressivity and flexibility in model design. Despite these advantages, computing likelihoods for CNF-generated samples requires expensive Jacobian trace evaluations along the integration path [13, 14]. This computational overhead limits their scalability, particularly for large-scale protein systems. In this work, we ask the question:

*Can the likelihood of large-scale scientific generative models be efficiently amortized to avoid the prohibitive path integral?*

Here, as a means to learn such a likelihood surrogate, we investigate the use of energy-based models (EBMs). EBMs learn the energy function of a synthetic Boltzmann distribution, $p_\theta(x) \propto \exp(E_\theta(x))$ where $E_\theta$ is the energy to be learned [15]. Scalable training of EBMs remains a major challenge due to the need for sampling from the model distribution, which often requires simulation during training [16, 17]. As a result, developing efficient training algorithms for EBMs continues to be an active area of research [15, 18–20].

We adopt noise contrastive estimation (NCE) [21], which trains a classifier to distinguish between samples drawn from the target distribution and those from a carefully chosen noise distribution. The key advantage of this approach is that it circumvents the need to compute intractable normalizing constants [22, 23]. Despite this, NCE can suffer from the *density-chasm* problem [24, 25], whereby the optimization landscape becomes flat when the data and noise distributions differ significantly [26]. To address this issue, we introduce an annealing scheme between a simple noise distribution and the data distribution using stochastic interpolants [27]. Annealing mitigates the density chasm problem by introducing intermediate distributions between noise and data, which facilitates the generation of more informative samples. In particular, it ensures that negative samples lie closer to positive samples, improving the effectiveness of NCE optimization [24, 28]. We further enhance the training process by combining an InfoNCE [29] loss with a score matching [30] objective defined over the law of the

stochastic interpolant. Notably, our proposed method for training the EBM is both *simulation-free* and avoids the computation of normalizing constants, making it scalable to large systems.

**Contributions.** On synthetic 2D systems, we show that training with both losses performs significantly better than either individually. For the alanine dipeptide system, our method recovers the correct semi-empirical free energy surface while achieving a $100\times$ speedup over exact likelihood calculations. On other dipeptide systems, we further demonstrate the method's ability to generalize to previously unseen molecular systems. To summarize, our *main contributions* are:

- **Training.** We develop a scalable, simulation-free framework for training EBMs by combining stochastic interpolants, score matching, and noise contrastive estimation.

- **Fast likelihoods.** We show that learned likelihoods can replace expensive Jacobian computations in the reweighting step, recovering exact Boltzmann statistics.

- **Empirical validation.** We achieve $100\times$ speedup on alanine dipeptide compared to exact likelihoods, and demonstrate $6\times$ speedup over MD on unseen dipeptide systems.

## 2 Methodological Framework

In this work, we introduce a new class of generative models for sampling from a Boltzmann distribution that features accelerated likelihood computations (Figure 1). To this end, we train a standard Boltzmann Emulator and an EBM, which each enable efficient sampling and likelihood evaluation, respectively. Given access to an EBM approximating the output distribution of the Boltzmann Emulator, we can evaluate sample likelihoods in just a single function call, enabling rapid reweighting for accurate estimation of observables. We train these EBM models with a new hybrid approach that we call *BoltzNCE*. As critical components of our approach, we first provide background on stochastic interpolants and flow matching [27, 31], then show how these can be used to build Boltzmann Emulators and finally, we introduce the innovations underlying the BoltzNCE method.

### 2.1 Background: stochastic interpolants and Boltzmann emulators

A stochastic interpolant [27, 32] is a stochastic process that smoothly deforms data from a fixed base distribution $\rho_0$ into data sampled from the target distribution $\rho_1 = p^*$. Under specific choices of the hyperparameters, stochastic interpolants recover standard settings of diffusion models [30, 33], flow matching [31], and rectified flows [34]. They can be used to learn generative models, because they provide access to time-dependent samples along a dynamical process that converts samples from the base into samples from the target. Concretely, given samples $\{x_1^i\}_{i=1}^n$ with $x_1^i \sim \rho_1$ sampled from the target, we may define a stochastic interpolant as the time-dependent process

$$I_t = \alpha_t x_0 + \beta_t x_1 \tag{1}$$

Above, $\alpha : [0, 1] \to \mathbb{R}$ and $\beta : [0, 1] \to \mathbb{R}$ are continuously differentiable functions satisfying the boundary conditions $\alpha_0 = 1, \alpha_1 = 0, \beta_0 = 0$, and $\beta_1 = 1$. In the absence of domain knowledge, we often take the base distribution to be a standard Gaussian, $\rho_0 = \mathsf{N}(0, I)$.

The probability density $\rho_t = \mathsf{Law}(I_t)$ induced by the interpolant coincides with the density of a probability flow that pushes samples from $\rho_0$ onto $\rho_1$,

$$\dot{x}_t = b_t(x_t), \qquad b_t(x) = \mathbb{E}[\dot{I}_t \mid I_t = x], \tag{2}$$

where $b_t(x)$ is given by the conditional expectation of the time derivative of the interpolant at a fixed point in space. The score $s_t(x) = \nabla \log \rho_t(x)$ is further given by the conditional expectation [27]:

$$s_t(x) = \alpha_t^{-1} \mathbb{E}\left[x_0 \mid I_t = x\right], \tag{3}$$

which we will use to train our energy-based model as a likelihood surrogate.

**Flow matching.** Given a coupling $\rho(x_0, x_1)$ between $\rho_0$ and $\rho_1$, the vector field (2) can be approximated with a neural network $\hat{b}$ via the regression problem

$$\mathcal{L}_b(\hat{b}) = \mathbb{E}\left[\|\hat{b}_t(I_t) - \dot{I}_t\|^2\right], \tag{4}$$

where $\mathbb{E}$ denotes an expectation over $(t, x_0, x_1)$. The objective (4) can be further modified so that the model is trained to predict the clean data $x_1$ instead of the vector field $\hat{b}$ [35, 36]. We refer the reader to Appendix A for more details on the endpoint objective.

Once trained, the learned model can be used to generate samples $\tilde{x}_0$ by solving the differential equation

$$\dot{\hat{x}}_t = \hat{b}_t(\hat{x}_t), \qquad x_0 \sim \rho_0. \tag{5}$$

The log density $\log \hat{\rho}$ associated with the generated samples $\hat{x}_1$ can be calculated with the continuous change of variables formula,

$$\log \hat{\rho}(\hat{x}_1) = \log \rho_0(x_0) - \int_0^1 \nabla \cdot \hat{b}_t(\hat{x}_t) dt. \tag{6}$$

While (6) gives a formula for the exact likelihood, its computation is expensive due to the appearance of the divergence of the estimated flow $\hat{b}$.

**Boltzmann Generators.** Boltzmann Generators leverage generative models to sample conformers and to compute their likelihoods, which enables reweighting the generated samples to the Boltzmann distribution. In practice, these models can be built using the stochastic interpolant framework described in (1), (2) and (4) given target data generated via molecular dynamics. We can generate unbiased samples from the target distribution by first sampling $\hat{x}_1 \sim \hat{\rho}_1$ by solving (5) and (6), and then by reweighting with the importance weight $w(\hat{x}_1) = \exp(\frac{-E(\hat{x}_1)}{K_B T})/\hat{\rho}_1(\hat{x}_1)$. With this weight, we can also approximate any observable $O$ under the Boltzmann distribution $\mu$ using self-normalized importance sampling [7, 8, 12]:

$$\langle O \rangle_\mu = \mathbb{E}_{\hat{x}_1 \sim \hat{\rho}_1} [w(\hat{x}_1) O(\hat{x}_1)] \approx \frac{\sum_{i=1}^N w(\hat{x}_1^i) O(\hat{x}_1^i)}{\sum_{i=1}^N w(\hat{x}_1^i)}. \tag{7}$$

While the likelihood integral (6) has been used in prior implementations of Boltzmann generators [8, 12], in general its computational expense prevents it from scaling to large molecular systems. In this work, our aim is to amortize the associated cost by learning a second energy-based model that can estimate the likelihoods $\log \hat{\rho}_1$.

## 2.2 BoltzNCE

Our method is designed to calculate free energies and to enable the computation of observables via (7) in an efficient and scalable manner. It proceeds in two stages: (i) standard flow training, and (ii) amortization of the likelihood via EBM training. In the first stage, we train a Boltzmann emulator on a dataset $\mathcal{D}$ of conformers using the stochastic interpolant framework described in Section 2.1. The trained emulator is then used to generate samples $\hat{x}_1 \sim \hat{\rho}_1$ via (5), leading to a dataset of generated conformers $\hat{\mathcal{D}}$. In the second stage, we train an EBM on $\hat{\mathcal{D}}$ to approximate the energy $U$ of the generated distribution,

$$\hat{\rho}_1(x) = \exp(U(x))/Z, \qquad Z = \int \exp(U(x)) dx. \tag{8}$$

The generated samples are then reweighted to the Boltzmann distribution using (8) in (7).

**Training the emulator.** We train the Boltzmann Emulator using stochastic interpolants as described in Section 2.1. Boltzmann Emulator models are trained using either the vector field (4) or the endpoint objectives. The *endpoint parameterization* [35, 36] is where the network predicts the clean endpoint $x_1$ rather than the velocity field directly. The corresponding velocity field for sampling is given by

$$\hat{b}_t(x) = \alpha_t^{-1}(\dot{\alpha}_t x + (\dot{\beta}_t \alpha_t - \beta_t)\hat{x}_1(t, x)) \tag{9}$$

where $\hat{x}_1(t, x)$ is the network's predicted endpoint at time $t$ given noisy sample $x$. A derivation of this velocity field is provided in Appendix A. Since $\alpha_1 = 0$, this velocity field diverges at $t = 1$; in practice, we integrate from $t = 0$ to $t = 1 - 1e^{-3}$ to avoid this singularity. While the endpoint parameterization works well in generating samples, it leads to unstable likelihoods due to the singularity at t=1, a drawback that is addressed by the use of EBMs in the BoltzNCE method. Once trained, we generate the dataset $\hat{\mathcal{D}}$ by sampling $\hat{x}_1 \sim \hat{\rho}_1$ via (5).

**Training the EBM.** Rather than learn a single energy function corresponding to $\hat{\rho}_1$ as in (8), we propose to define a second stochastic interpolant from the base to $\hat{\rho}_1$,

$$\tilde{I}_t = \alpha_t x_0 + \beta_t \hat{x}_1, \tag{10}$$

and estimate the associated *time-dependent* energy function $U_t$ for $\tilde{\rho}_t = \mathsf{Law}(\tilde{I}_t)$,

$$\tilde{\rho}_t(x) = \exp\left(U_t(x)\right)/Z_t, \qquad Z_t = \int \exp(U_t(x))dx. \tag{11}$$

By construction, we have that $\tilde{\rho}_1 = \hat{\rho}_1$, though in general $\tilde{\rho}_t \neq \hat{\rho}_t = \mathsf{Law}(\hat{x}_t)$. Given access to a model $\hat{U}_t$ of $U_t$, we can evaluate $\log\hat{\rho}_1(x)$ up to the normalization constant $\hat{Z}_1$ with a single function evaluation $\hat{U}_1(x)$, eliminating the need for (6). We train $\hat{U}$ using a combination of denoising score matching and an InfoNCE objective [29]. The score matching objective leverages (3) to yield

$$\mathcal{L}_{\mathrm{SM}}(\hat{U}) = \mathbb{E}\left[|\alpha_t \nabla \hat{U}_t(\tilde{I}_t) + x_0|^2\right], \tag{12}$$

where the $\mathbb{E}$ is over the draws of $(t, x_0, \hat{x}_1)$ defining $\tilde{I}_t$. To define the InfoNCE objective, we first write down the joint distribution over $(t, \tilde{I}_t)$ given by $\tilde{\rho}(t, x) = p(t)\tilde{\rho}_t(x)$ for $t \sim p(t)$. In practice, we choose time uniformly, so that $p(t) = 1$ and $\tilde{\rho}(t, x) = \tilde{\rho}_t(x)$. Given this joint density, we may define the conditional distribution $\tilde{\rho}(t|x) = \tilde{\rho}(t, x)/\tilde{\rho}(x) = \tilde{\rho}_t(x)/\tilde{\rho}(x)$ where $\tilde{\rho}(x) = \int \tilde{\rho}(t, x)dt = \int \tilde{\rho}_t(x)dt$ is the marginal distribution of $x$. This conditional distribution describes the probability that a given observation $x$ was a sample at time $t$. The InfoNCE objective maximizes the conditional likelihood by minimizing its negative log-likelihood. We can write this intractable quantity as

$$\mathrm{NLL} = -\mathbb{E}\left[\log\left(\frac{\tilde{\rho}_t(x)}{\tilde{\rho}(x)}\right)\right] \approx -\mathbb{E}\left[\log\left(\frac{\exp(\hat{U}_t(\tilde{I}_t) - \log\hat{Z}_t)}{\int \exp(\hat{U}_{t'}(\tilde{I}_t) - \log\hat{Z}_{t'})dt'}\right)\right] = \mathcal{L}_{\mathrm{NLL}}(\hat{U}), \tag{13}$$

where the expectation is over the draws of $(t, x_0, \hat{x}_1)$ defining $\tilde{I}_t$. This expression depends on unknown normalization constants $\hat{Z}_t$ and involves an integral over time $t'$. To avoid explicit normalization, we parameterize $\hat{U}_t$ to directly approximate $U_t(x) - \log Z_t$, absorbing the log-normalization constant as a time-dependent bias. To approximate the integral, InfoNCE leverages a Monte Carlo approximation, leading to a multi-class classification problem: given a sample $\tilde{I}_t$, we aim to distinguish the true time $t$ from a set of negative times $\{\tilde{t}_k\}_{k=1}^K$. This yields the tractable objective:

$$\mathcal{L}_{\mathrm{InfoNCE}}(\hat{U}) = -\mathbb{E}\left[\log\left(\frac{\exp(\hat{U}_t(\tilde{I}_t))}{\sum_{t' \in \{\tilde{t}_k\} \cup \{t\}} \exp(\hat{U}_{t'}(\tilde{I}_t))}\right)\right]. \tag{14}$$

We train $\hat{U}$ by minimizing a weighted combination of both objectives:

$$\mathcal{L}_{\mathrm{BoltzNCE}}(\hat{U}) = \mathcal{L}_{\mathrm{SM}}(\hat{U}) + \mathcal{L}_{\mathrm{InfoNCE}}(\hat{U}) \tag{15}$$

Since the noise level changes continuously with $t$, only nearby times $t' \approx t$ have non-negligible conditional likelihood $\tilde{\rho}(t'|\tilde{I}_t)$. We therefore sample negatives $\{\tilde{t}_k\}$ from a narrow Gaussian centered at $t$, providing informative contrast for learning fine-grained temporal discrimination. We note that both objective functions are *simulation-free* after generation of $\hat{\mathcal{D}}$, as they only require sampling the interpolant $\tilde{I}_t$. While, in principle, the EBM could be trained on the original interpolant $I_t$ (1), training on the generated interpolant $\tilde{I}_t$ (10) is preferable because it approximates the emulator's energy function and enhances transferability to new molecular systems where long-timescale MD data is unavailable.

For more details on model training and inference, refer to Appendices B, E.8 and E.10.

## 3 Related Work

**Boltzmann Generators.** Boltzmann Generators have become an active line of research since their initial development with invertible neural networks [7], having now been used to sample both molecular systems [10, 11, 37–40] and lattice systems [10, 41–43]. Tan et al. [44] introduce a more stable reweighting scheme that takes advantage of Jarzynski's equality to attain the equilibrium distribution.

However, most of these methods have required input through system-specific featurizations such as internal coordinates, hindering transferability. The emergence of CNFs and equivariant neural network architectures has enabled the development of BGs on Cartesian coordinates [8, 12] thereby not being system specific and enabling transferability. Despite these advancements, transferability has so far only been demonstrated on small systems such as dipeptides, primarily due to the computational limitations associated with likelihood evaluation at scale.

**Boltzmann Emulators.** Boltzmann Emulators, unlike BGs, are designed solely to produce high-quality samples without reweighting to the Boltzmann distribution. Because they are not required to be invertible, they can typically be applied to much larger systems. This flexibility also enables the use of a wider range of generative approaches, including diffusion models. Boltzmann Emulators have been employed to generate peptide ensembles [45], protein conformer distributions [35, 46–49], small molecules [50–52], and coarse-grained protein structures [53, 54]. However, they are inherently limited by the data distribution they were trained on. As a result, they are generally unsuitable for generating unbiased samples from the Boltzmann distribution or for performing free energy calculations independently. In this work, we aim to leverage the strengths of Boltzmann Emulators and bridge the gap between Emulators and Generators using energy-based models (EBMs). While EBMs may still yield biased estimates, they bring us closer to true Boltzmann statistics at a fraction of the computational cost.

**Energy-Based Models.** Energy-Based Models (EBMs) are particularly appealing in the physical sciences, as they describe density functions in a manner analogous to the Boltzmann distribution. This similarity enables the use of various techniques from statistical physics to compute thermodynamic properties of interest [39, 55]. Despite their promise, training EBMs remains a challenging task. Recent advancements have introduced training objectives inspired by noise contrastive estimation [18, 20, 24, 25, 55, 56], contrastive learning [29, 57], and score matching [15, 58, 59]. OuYang et al. [60] have also proposed an "energy-matching" objective to train a neural sampler on the Boltzmann distribution; however, more work needs to be done to make this approach practical for molecules.

# 4 Experiments

In this section, we validate BoltzNCE on several synthetic low-dimensional systems as well as on molecular conformer generation for dipeptides. Our experiments demonstrate three key results: (i) Combining the denoising score matching (12) and InfoNCE (14) objectives to form (15) significantly improves EBM quality compared to using either alone; (ii) BoltzNCE enables endpoint-parameterized emulators to function as Boltzmann Generators by decoupling likelihood estimation from the sampling vector field, avoiding the numerical instability that makes Jacobian trace computation intractable; (iii) The learned EBM provides accurate and efficient likelihood estimates, enabling the calculation of Boltzmann statistics at speeds up to two orders of magnitude faster than exact likelihood computations.

## 4.1 Low-dimensional systems

The effectiveness of the score matching and InfoNCE loss functions (12) and (14) are tested on low-dimensional synthetic systems. Specifically, we study an 8-mode Gaussian mixture in two dimensions and the two-dimensional checkerboard distribution. The results for forward evaluation of the trained EBMs are shown in Figure 2. We leverage a simple feedforward neural network that takes the point $x \in \mathbb{R}^2$ as input and outputs a scalar value. KL-divergence values between the predicted and ground truth densities are also reported in Table 1.

| Objective Functions | 8-Mode Gaussian Mixture | Checkerboard |
|---|---|---|
| InfoNCE | 0.2395 | 3.8478 |
| Score Matching | 0.2199 | 0.8384 |
| Score Matching & InfoNCE | **0.0150** | **0.1987** |

Table 1: **Low-dimensional Systems: Quantitative Results.** KL-divergence (↓) when using different objective functions to train the EBM. For both systems, using our combined objective (15) significantly outperforms using either (12) or (14) individually.

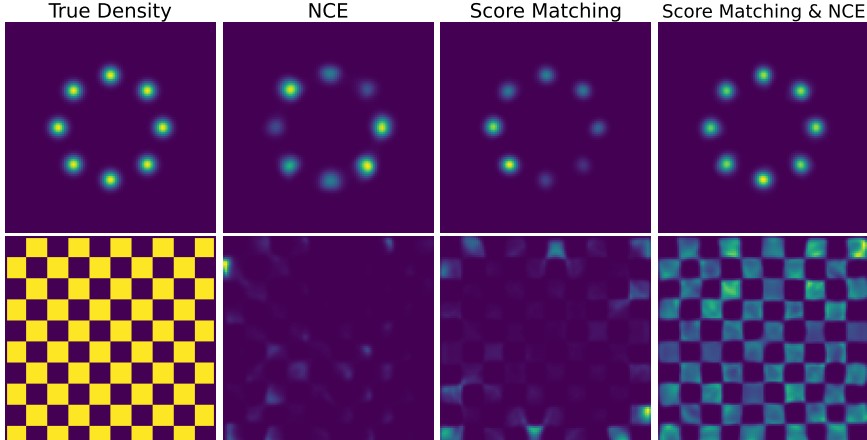

Figure 2: **Low-dimensional Systems: Qualitative Results.** EBM density learned on synthetic two-dimensional systems. (Above) An 8-mode Gaussian mixture. (Below) The checkerboard distribution. In both cases, the true density is shown in the leftmost column, and the results obtained with different methods are shown to the right. Using both objectives (right) provides the best performance.

Our results indicate that combining both objective functions leads to significantly improved performance compared to using either alone. Intuitively, the InfoNCE loss appears to act as a regularizer for the score matching objective, helping to improve the relative weighting of different modes in the learned distribution. This effect is not observed when using the score matching loss on its own. For subsequent experiments on molecules, both objective functions are used to train the EBM.

## 4.2 Alanine Dipeptide

As a more complex system, we first study the alanine dipeptide molecule. Our aim here is to obtain the equilibrium distribution of the molecule as specified by the semi-empirical GFN-xTB force field [61]. Running a simulation with this force field is computationally intensive, so we use the same setup as Klein and Noé [8]. Conformers are generated using molecular dynamics with the classical Amber ff99SBildn force field and subsequently relaxed using GFN-xTB. We use two dataset variants: *unbiased*, corresponding to the original distribution, and *biased*, in which the positive $\varphi$ state is oversampled for equal metastable state representation (Figure 6).

We train geometric vector perceptron (GVP)[62, 63] based Boltzmann Emulators using both the vector field objective (4) (GVP-VF) and the endpoint objective (GVP-EP) on the unbiased and biased datasets, and compare their performance to Equivariant Continuous Normalizing Flow (ECNF) models from Klein and Noé [8] trained on the same datasets. Additionally, we use the Graphormer architecture [64] to parameterize the EBM. Further details on data featurization, model architectures, and hyperparameters are provided in Appendices C and E.6.

Table 2: **Methods Overview** Flow matching objectives and likelihood estimation methods for different models tested on Alanine Dipeptide.

| Method | FM Objective | Likelihood Estimation |
|---|---|---|
| ECNF [8] | Vector Field | Jac-trace integral |
| GVP Vector field | Vector Field | Jac-trace integral |
| GVP Endpoint | Endpoint | Jac-trace integral |
| BoltzNCE Vector field | Vector Field | **EBM forward pass** |
| BoltzNCE Endpoint | Endpoint | **EBM forward pass** |

To evaluate Boltzmann generation, we use models trained on the biased dataset, as these yield more accurate estimates of free energy differences. Free energy differences between the positive and negative $\varphi$ metastable states are computed, since this transition corresponds to a slow dynamical process (Appendix D.4, Figure 6). Ground-truth free energy values are obtained via umbrella sampling simulations from Klein et al. [12]; further details on umbrella sampling are provided in Appendix E.2. In addition, we compute the energy-based ($\mathcal{E}$–$W_2$) and torsion-based ($\mathbb{T}$–$W_2$) Wasserstein-2 distances between the unbiased dataset and the generated (proposal or reweighted) distributions.

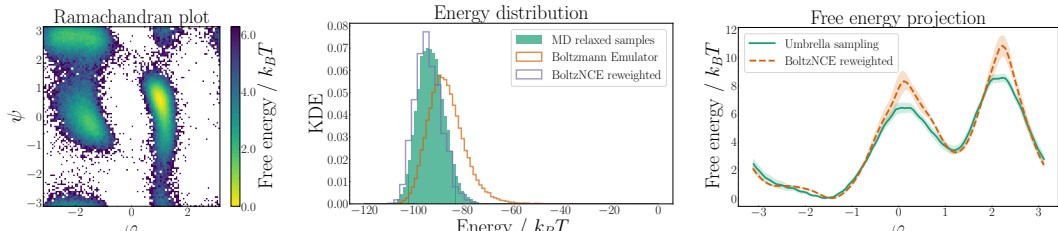

Figure 3: **BoltzNCE: Qualitative Results.** Results for BoltzNCE on alanine dipeptide trained on the biased dataset. We use a GVP vector field as the Boltzmann Emulator. BoltzNCE successfully captures the energy distribution and the free energy projection. (Left) Ramachandran plot of generated samples. (Middle) Energy histogram along with BoltzNCE reweighting. (Right) Calculated free energy surfaces for the angle $\varphi$ on the right.

| Method | $\Delta F/k_B T$ | $\Delta F$ Err. | Prop. $\mathcal{E}$-$W_2$ | Rew. $\mathcal{E}$-$W_2$ | Prop. $\mathbb{T}$-$W_2$ | Rew. $\mathbb{T}$-$W_2$ | Inf. (h) | Train (h) |
|---|---|---|---|---|---|---|---|---|
| Umbrella Sampling | $4.10 \pm 0.26$ | – | – | – | – | – | – | – |
| ECNF [8] | $\mathbf{4.09 \pm 0.05}$ | $\mathbf{0.01 \pm 0.05}$ | – | – | – | – | 9.37 | $\mathbf{3.85}$ |
| ECNF (reproduced) | $4.07 \pm 0.23$ | $0.03 \pm 0.23$ | $8.08 \pm 0.56$ | $0.37 \pm 0.02$ | $1.10 \pm 0.01$ | $0.59 \pm 0.00$ | 9.37 | $\mathbf{3.85}$ |
| GVP Endpoint | $4.89 \pm 2.61$ | $0.79 \pm 2.61$ | $\mathbf{6.19 \pm 1.08}$ | $2.88 \pm 0.01$ | $1.12 \pm 0.01$ | $0.58 \pm 0.01$ | 26.2 | 4.42 |
| GVP Vector Field | $4.38 \pm 0.67$ | $0.28 \pm 0.67$ | $7.20 \pm 0.13$ | $0.46 \pm 0.05$ | $\mathbf{1.09 \pm 0.01}$ | $0.60 \pm 0.00$ | 18.4 | 4.42 |
| BoltzNCE Endpoint | $4.14 \pm 0.94$ | $0.04 \pm 0.94$ | $6.24 \pm 0.52$ | $2.78 \pm 0.04$ | $1.12 \pm 0.01$ | $0.59 \pm 0.01$ | 0.16 | 12.2 |
| BoltzNCE Vector Field | $4.08 \pm 0.13$ | $0.02 \pm 0.13$ | $7.12 \pm 0.15$ | $\mathbf{0.27 \pm 0.02}$ | $1.12 \pm 0.00$ | $\mathbf{0.57 \pm 0.00}$ | $\mathbf{0.09}$ | 12.2 |

Table 3: **BoltzNCE: Quantitative Results.** Dimensionless free energy difference, energy $\mathcal{E}$-$W_2$ and torsion angle $\mathbb{T}$-$W_2$ Wasserstein-2 distances calculated by different Boltzmann Generator and BoltzNCE models. Standard deviations are shown across 5 runs. Free energy difference values for umbrella sampling and ECNF taken from Klein and Noé [8], which we consider to be ground truth. BoltzNCE Vector Field provides the best performance/inference time tradeoff as compared to all other methods.

The ECNF, GVP-VF, and GVP-EP models estimate likelihoods using the Jacobian trace integral and act as Boltzmann Generators. Energy-based models (EBMs) trained on samples generated by the GVP-based emulators are also evaluated and referred to as BoltzNCE-VF and BoltzNCE-EP. The specific flow-matching objectives and likelihood estimation methods for each model are summarized in Table 2.

**GVP models excel as Emulators but fail as Generators due to poor likelihood estimates.** The results for the Boltzmann Emulator models trained on the unbiased dataset are given in Appendix F.1. In general, the two GVP models demonstrate better performance than the ECNF. The GVP-EP model performs the best, making it a strong candidate for a Boltzmann Emulator, however, as shown below, it fails as a Boltzmann Generator as its vector field (Eq. 9) diverges as $t \to 1$, which we find corrupts its likelihood calculation.

All Boltzmann generation results are summarized in Table 3. Comparing the GVP and ECNF models, we find that while GVP models match or exceed ECNF performance as emulators, they produce less accurate free energy differences and exhibit higher reweighted $\mathcal{E}$–$W_2$ and $\mathbb{T}$–$W_2$ scores. This inaccuracy may stem from unreliable likelihood estimates produced during ODE integration, which requires the divergence of the model with respect its input to be accurate and well-behaved. As a result, Boltzmann Generators face additional design constraints to ensure stability and reliability of their likelihood computation. In the following, we show how BoltzNCE resolves this issue by learning likelihoods separately, decoupling emulator quality from likelihood tractability and enabling greater flexibility in the design of the emulator.

**BoltzNCE enables fast and accurate likelihood estimation for Emulators.** In contrast, the BoltzNCE models yield more accurate estimates of the free energy difference and achieve lower reweighted $\mathcal{E}$-$W_2$ and $\mathbb{T}$-$W_2$ scores, with BoltzNCE-VF providing the best performance. This indicates that the likelihoods predicted by BoltzNCE are generally more reliable than those obtained via the Jacobian trace integral. Representative energy histograms and free energy surfaces along the slowest transition ($\varphi$ dihedral angle) for the BoltzNCE-VF model are shown in Figure 3. For energy histograms and free energy projections of other methods, refer to Appendix F. As demonstrated in

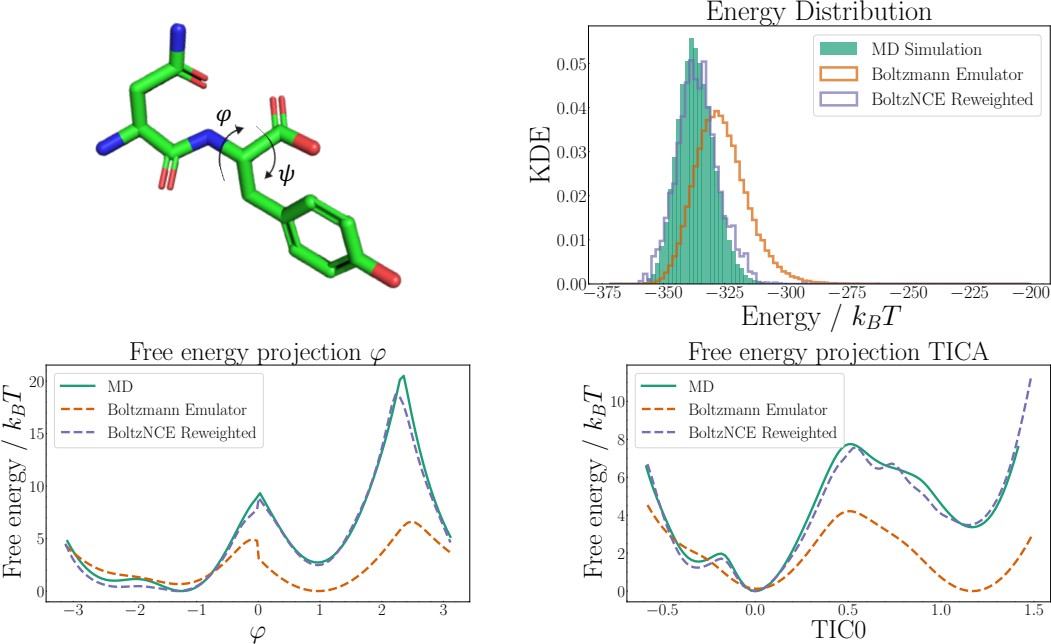

Figure 4: **BoltzNCE Results on NY dipeptide.** BoltzNCE inference results for NY dipeptide (top left) after fine-tuning. Energy distribution (top right), free energy surfaces along the $\varphi$ angle (bottom left) and the first TICA component (bottom right). BoltzNCE successfully captures the right energy distribution and free energy projections for the dipeptide.

the figure, the BoltzNCE method is able to accurately capture the free energy projection and energy distribution for alanine dipeptide.

We report inference time costs in Table 3, including the time to generate and estimate likelihoods for $10^6$ conformers. BoltzNCE provides an overwhelming inference time advantage over the standard Boltzmann Generator by two orders of magnitude while matching accuracy.

To further evaluate the accuracy of EBM likelihoods, we show that it is both more accurate and more computationally efficient than the Hutchinson trace estimator (Appendix F.3). We also demonstrate further flexibility in the design of the EBM training algorithm by comparing OT to independent coupling (Appendix F.4). It is important to note, however, that BoltzNCE has an upfront cost of training the EBM associated with it. In principle, this upfront cost could be reduced by training the EBM in parallel on the original interpolant (1) and then fine-tuning on the generated interpolant (10).

### 4.3 Generalizability on dipeptides

Finally, we demonstrate BoltzNCE's ability to generalize to unseen molecules using systems of dipeptides. For this experiment, we use the same setup and dataset from Klein and Noé [8], which was originally developed in Klein et al. [65]. The training set consists of classical force field MD simulations of 200 dipeptides, each run for about 50 ns. Since this dataset may not have reached convergence, we bias the dataset in a similar manner to alanine dipeptide to ensure equal representation of the modes along the $\varphi$ angle. For testing, we utilize 1 $\mu$s long simulations of 7 randomly chosen dipeptides not present in the training set.

In this experiment, we benchmark BoltzNCE against independent $1\mu s$ MD runs and the TBG-ECNF model [8] trained on the biased dataset. We use the BoltzNCE-VF method as it achieved the best performance on the alanine dipeptide system. We also compute time-lagged independent components (TICA) [66] from the test MD simulations and plot the free energy projections along the first component. For details on TICA, refer to Appendix E.5.

The inference procedure with BoltzNCE is modified to improve generalizability. Samples generated from the flow-matching model are passed through a conformer matching and chirality checking procedure to check validity of generated samples. For more details, refer to Appendices E.3 and E.4. The EBM, on the other hand, is first pretrained using conformers of training peptides generated by the

| Method | $\varphi \, \Delta F$ Error | $\mathcal{E}\text{-}W_2$ | $\mathbb{T}\text{-}W_2$ | Inference Time (h) |
|---|---|---|---|---|
| MD Baseline | $0.18 \pm 0.22$ | $\mathbf{0.17 \pm 0.03}$ | $\mathbf{0.22 \pm 0.03}$ | 24.04 |
| TBG-ECNF* [8] | $\mathbf{0.13 \pm 0.10}$ | $0.36 \pm 0.12$ | $0.34 \pm 0.06$ | 123.07 |
| BoltzNCE | $0.43 \pm 0.21$ | $1.08 \pm 0.65$ | $0.44 \pm 0.13$ | **4.005** |

Table 4: **Generalizability Results.** Generalizability of the BoltzNCE method in comparison to TBG and MD simulations on systems of 7 dipeptides. *Fewer samples (30,000) used due to high GPU compute time. BoltzNCE provides a significant time advantage over the other methods while achieving good performance.

flow-matching model and then fine-tuned on the dipeptide of interest during inference. In addition, we exclude the top 0.2% of importance weights during reweighting to reduce variance, following the approach introduced in Tan et al. [44].

**BoltzNCE yields accurate Boltzmann statistics at a fraction of MD/TBG computational cost.** Quantitative evaluations across seven dipeptide systems (Table 4) show that BoltzNCE closely reproduces Boltzmann-weighted energy distributions and free energy surfaces obtained from molecular dynamics (MD), as illustrated for the NY dipeptide in Figure 4. Although the TBG-ECNF method attains the highest accuracy in free energy estimation, it incurs orders-of-magnitude higher computational cost due to its reliance on exact likelihood calculations, thereby serving as an upper bound on BoltzNCE performance. In contrast, BoltzNCE achieves comparable accuracy by approximating likelihoods at substantially lower computational expense.

Visual inspection of all test systems (Figure 10) confirms that this minor performance drop remains acceptable, with BoltzNCE exhibiting excellent agreement with MD-derived energy distributions and free energy landscapes. A small reduction in $\mathcal{E}\text{-}W_2$ scores, primarily driven by a single outlier in the NF dipeptide (Appendix G), does not affect overall fidelity. Collectively, these results demonstrate that BoltzNCE provides an efficient, scalable, and generalizable framework for amortized Boltzmann sampling on unseen molecular systems, maintaining high thermodynamic accuracy at a fraction of the computational cost.

## 5    Discussion

In this work, we introduce a novel, scalable, and simulation-free framework for training energy-based models that integrates stochastic interpolants, InfoNCE, and score matching. We show that InfoNCE and score matching act complementarily to enhance model performance. Our approach learns the density of conformers sampled from a Boltzmann Emulator, eliminating the need for costly Jacobian trace calculations and achieving orders-of-magnitude speedups. On alanine dipeptide, BoltzNCE can even surpass the accuracy of ODE-based divergence integration. Across multiple dipeptide systems, the method generalizes to unseen molecules with minimal fine-tuning, providing substantial computational savings over conventional molecular dynamics. This framework bridges the gap between Boltzmann Emulators and Generators, removing the dependence on invertible architectures and expensive likelihood computations, while enabling high-fidelity, scalable Boltzmann sampling.

## 6    Limitations and Future Work

The present work is limited to dipeptide molecular systems. While on ADP the method demonstrates excellent accuracy, the performance drops in the generalizability settings. This limitation could likely be addressed through more extensive exploration of model architectures and hyperparameter optimization, which we have only minimally investigated here. The method also has the potential to be scaled to larger molecular systems; however, the accuracy of the method needs to be further tested in higher-dimensional settings.

Training the energy-based model requires applying the score matching loss to its gradients, which increases compute requirements beyond typical levels for neural networks training. Additionally, since the likelihoods estimated by the EBM are approximate, a degree of mismatch between the samples and their predicted likelihoods is inevitable.

Although the current work is limited to a molecular setting, we believe the proposed EBM training framework could be broadly applicable in other domains where energy-based models are useful, such as inverse problems in computer vision and robotics.

# 7 Acknowledgments

We thank Leon Klein for making the code and data for his original Boltzmann Generator and Equivariant Flow Matching methods readily available.

This work is funded through R35GM140753 from the National Institute of General Medical Sciences. The content is solely the responsibility of the authors and does not necessarily represent the official views of the National Institute of General Medical Sciences or the National Institutes of Health.

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

# A Endpoint Objective

Stochastic interpolants anneal between $x_0 \sim \mathcal{N}(0, \mathbf{I})$ and $x_1 \sim p_*(x)$ with:

$$I_t = \alpha_t x_0 + \beta_t x_1 \tag{16}$$

solving for $x_0$:

$$x_0 = \frac{I_t - \beta_t x_1}{\alpha_t} \tag{17}$$

$I_t$ evolves according to the vector field given by the conditional expectation:

$$b_t(x) = \dot{\alpha}_t \mathbb{E}\left[x_0 | I_t = x\right] + \dot{\beta}_t \mathbb{E}\left[x_1 | I_t = x\right] \tag{18}$$

Substituting 17 in 18 we get:

$$b_t(x) = \frac{\dot{\alpha}_t (x - \beta_t \mathbb{E}\left[x_1 | I_t = x\right])}{\alpha_t} + \dot{\beta}_t \mathbb{E}\left[x_1 | I_t = x\right] \tag{19}$$

$$b_t(x) = \alpha_t^{-1}(\dot{\alpha}_t x + (\dot{\beta}_t \alpha_t - \beta_t)\mathbb{E}\left[x_1 | I_t = x\right]) \tag{20}$$

Similarly, the model estimate of the vector field is given by:

$$b_{\theta,t}(x) = \alpha_t^{-1}(\dot{\alpha}_t x + (\dot{\beta}_t \alpha_t - \beta_t)\hat{x}_1(t, x)) \tag{21}$$

Where $\hat{x}_1(t, I_t)$ is the predicted endpoint by the model. The objective is then given by:

$$\mathcal{L}_{EP} = \int_0^T \|b_{\theta,t}(I_t) - b_t(I_t)\|^2 dt \tag{22}$$

$$\mathcal{L}_{EP} = \int_0^T \mathbb{E}\left[\|\frac{\dot{\beta}_t \alpha_t - \beta_t}{\alpha_t}(\hat{x}_1(t, I_t) - x_1)\|^2\right] dt \tag{23}$$

# B EBM training algorithm

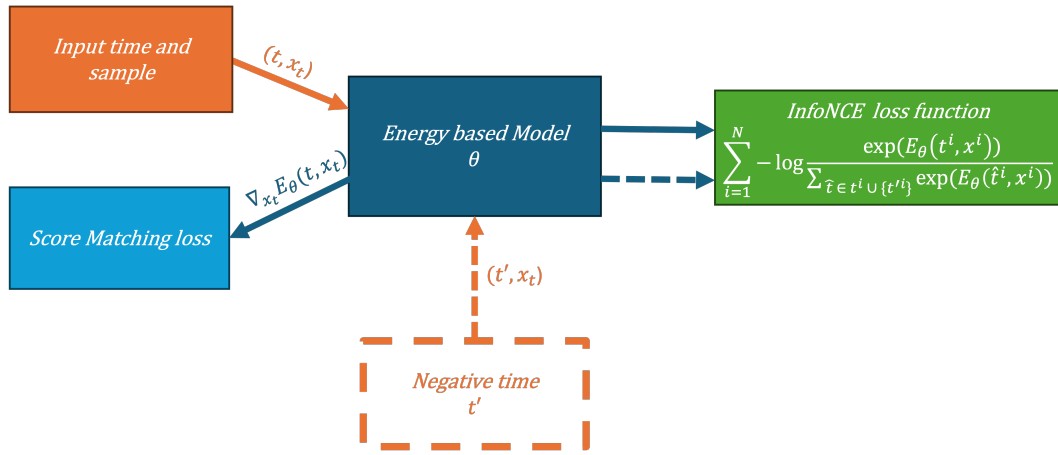

Figure 5: Energy Based Model training workflow

A diagrammatic representation of the method used for training the energy-based model is shown in Figure 5. The model takes a sample $x$ and time point $t$ as input and outputs predicted energy $E_\theta(t, x)$. The gradient of the output with respect to the sample $\nabla_x E_\theta(t, x)$ is used for the score matching loss. The same sample is also passed with negative time points $\{t'\}$, and the predicted energies $E_\theta(t', x)$ are used along with the previously output energies $E_\theta(t, x)$ for the InfoNCE loss.

We also provide a pseudocode block for training the energy-based model with stochastic interpolants, InfoNCE, and score matching in algorithm block 1.

---

**Algorithm 1:** Training EBM with stochastic interpolants, InfoNCE, and score matching

---

**Input:** Energy-Based model $\theta$, samples from prior $X_0$, generated samples $\tilde{X}_1$, interpolant functions $\alpha_t$, $\beta_t$, negative time sampling variance $\sigma$

**for** $epoch \leftarrow 1$ **to** $epoch_{\max}$ **do**

    **for** *batch* $(x_0, x_1)$ *in* $(X_0, \tilde{X}_1)$ **do**

        $(x_0, \tilde{x}_1) \leftarrow$ coupling function$(x_0, \tilde{x}_1)$

        **sample** $t \sim \mathcal{U}(0, 1)$

        $I_t \leftarrow \alpha_t x_0 + \beta_t \tilde{x}_1$

        $\mathcal{L}_{SM} \leftarrow \frac{1}{N} \sum_{n=1}^{N} |\alpha_t \nabla E_\theta(t^n, I_t^n) + x_0^n|^2$

        **sample** $t' \sim \mathcal{N}(t, \sigma^2)$

        $\mathcal{L}_{\text{InfoNCE}} \leftarrow \frac{1}{N} \sum_{n=1}^{N} -\log \frac{\exp(E_\theta(t^n, I_t^n))}{\exp(E_\theta(t^n, I_t^n)) + \exp(E_\theta(t'^n, I_t^n))}$

        $\mathcal{L} \leftarrow \mathcal{L}_{SM} + \mathcal{L}_{\text{InfoNCE}}$

        $\theta \leftarrow$ Update$(\theta, \nabla_\theta \mathcal{L})$

**Output:** Updated model parameters $\theta$

---

## C    Data featurization and Model Architecture

### C.1    Data featurization

The data is featurized such that all atom types are defined according to the peptide topology. For transferable models, amino acid identity and positional information are included along with atomic features. Molecular structures are represented as fully connected graphs, and both models operate directly on the Cartesian coordinates of the atoms.

### C.2    Geometric Vector Perceptrons

Boltzmann Emulators are parameterized with an SE(3)-equivariant graph neural network that leverages geometric vector perceptrons (GVPs) [62]. Briefly, a GVP maintains a set of equivariant vector and scalar features per node that is updated in an SE(3)-equivariant/invariant manner via graph convolutions. We utilize this architecture as it has been shown to have improved performance over equivariant graph neural networks (EGNNs) [67] in molecular design tasks [68].

For our models, we use a modified version of the GVP which has been shown to increase performance as described in Dunn and Koes [69]. The message passing step is constructed by applying the GVP message passing operation defined in Jing et al. [63].

$$(m_{i \to j}^{(s)}, m_{i \to j}^{(v)}) = \psi_M \left( [h_i^{(l)} : d_{ij}^{(l)}], \ v_i : \left[ \frac{x_i^{(l)} - x_j^{(l)}}{d_{ij}^{(l)}} \right] \right) \tag{24}$$

Here $m_{i \to j}^{(v)}$ and $m_{i \to j}^{(s)}$ are the vector and scalar messages between nodes $i, j$. $h_i$, $d_{ij}$ are the scalar features, edge features, and a radial basis embedding respectively, while $x$ represents the coordinates of the node. For the detailed Node Position Update and Node Feature Update operations, refer to Appendix C of Dunn and Koes [69].

## C.3 Graphormer Operations

Our EBMs are implemented using the graphormer [64] architecture, which has demonstrated state-of-the-art performance in molecular property prediction tasks. Graphormers function similarly to standard transformers, with the key difference being the incorporation of an attention bias derived from graph-specific features. In 3D-graphormers, this attention bias is computed by passing a Euclidean distance matrix through a multi-layer perceptron (MLP).

Graphformers are neural network architectures where layer-wise GNN components are nested alongside typical transformer blocks [70]. For our EBMs, we follow the implementation of the Graphformer with one minor modification. For the original Graphformer, each attention head is calculated as:

$$\text{head} = \text{softmax}\left(\frac{QK^\mathsf{T}}{\sqrt{d}} + B\right)V \tag{25}$$

where $B$ is a learnable bias matrix. In our implementation, $B$ is calculated by passing the graph's euclidean distance matrix through an MLP.

# D  Metrics

## D.1  NLL

To calculate the NLL of the holdout conformers, we take (6) and evaluate the ODE in the reverse direction for a given sample. This provides the NLL of the sample. NLL values are reported over batches of $1 * 10^3$ samples.

## D.2  Energy - W2

In order to quantify the difference in energy distributions between generated molecules and MD relaxed samples, we calculate the Wasserstein-2 distance between the two distributions. This can be intuitively thought of as the cost of transforming one distribution to another using optimal transport. Mathematically, we solve the optimization process with the loss:

$$\mathcal{E}\text{-}W_2 = \left(\inf_\pi \int c(x,y)^2 \, d\pi(x,y)\right)^{\frac{1}{2}} \tag{26}$$

where $\pi(x,y)$ represents a coupling between two pairs $(x,y)$ and $c(x,y)$ is the euclidean distance. We use the Python Optimal Transport package in our implementation [71]. $\mathcal{E}\text{-}W_2$ values are reported over batches of $1 * 10^5$ samples.

## D.3  Angle - W2

Similar to the $\mathcal{E}\text{-}W_2$ metric, we seek to quantify the differences in the distributions of dihedral angles generated and those from MD relaxed samples. Here, following the convention defined in Tan et al. [44], we define the optimal transport in torsional angle space as:

$$\mathbb{T}\text{-}W_2 = \left(\inf_\pi \int c(x,y)^2 \, d\pi(x,y)\right)^{\frac{1}{2}} \tag{27}$$

where $\pi(x,y)$ represents a coupling between two pairs $(x,y)$. The cost metric on torsional space is defined as:

$$c(x,y) = \left(\sum_{i=1}^{2s}\left((x_i - y_i)\%\pi\right)^2\right)^{\frac{1}{2}} \tag{28}$$

where $(x,y) \in [-\pi,\pi)^{2s}$

Similar to Energy-W2 calculations, we use the Python Optimal Transport package for implementation [71]. $\mathbb{T}\text{-}W_2$ values are computed in radians over batches of $1 \times 10^5$ samples.

### D.4 Free energy difference

We believe the free energy projection of a system is a relevant baseline for the following two reasons:

- **It represents a high dimensional integral of probability:** The equation of a free energy projection along a reaction coordinate is given by:

$$F(r') = -K_B T \ln \rho(r') \tag{29}$$

  where $\rho(r')$ is the probability density of observing the system at position $r'$ along the coordinate,

$$\rho(r') = \frac{1}{Z} \int_x \delta(r(x) - r') e^{(-U(x)/K_B T)} \, dx \tag{30}$$

  In principle, if we solve the above integral along all points of the reaction coordinate, we solve a $(D-1)$ dimensional integral, where $x \in R^D$. Thus, this serves as a good metric on how well the model matches the ground truth $R^D$ distribution.

  For dipeptides, the process of going from the negative $\varphi$ angle to the positive $\varphi$ angle is considered a slow process as there are regions of high energy/low probability in between the two. Therefore, this serves as an ideal reaction coordinate to study dipeptide systems.

- **Domain relevance:** Applied studies in the biophysical/biochemical/structural biology domain work on elucidating the free energy projection along a **reaction coordinate**. This helps the researchers identify the relative stability of different modes along the coordinate as well as the rate of reaction along the coordinate.

Free energy differences are computed between the positive and negative metastable states of the $\varphi$ dihedral angle. The positive state is defined as the region between 0 and 2, while the negative state encompasses the remaining range. The free energy associated with each state is estimated by taking the negative logarithm of the reweighted population count within that state.

The code for calculating the free energy difference is as follows:

```
left = 0.
right = 2

hist, edges = np.histogram(phi, bins=100, density=True,weights=weights)
centers = 0.5*(edges[1:] + edges[:-1])
centers_pos = (centers > left) & (centers < right)

free_energy_difference = -np.log(hist[centers_pos].sum()/
hist[~centers_pos].sum())
```

Where *phi* is a numpy array containing the $\varphi$ angles of the generated dataset ($\varphi \in (-\pi, \pi]$) and *weights* is an array containing the importance weight associated with it.

### D.5 Inference times

Inference time for free energy estimation is measured over $1 \times 10^6$ samples. Specifically, we use a batch size of 500 and generate 200 batches of conformers. During sample generation, Boltzmann Generators also compute the Jacobian trace. All run times are recorded on NVIDIA L40 GPUs. Reported values represent the mean of five independent runs for alanine dipeptide and the average across individual runs on the seven test-system dipeptides in the generalization setting.

## E    Technical Details

### E.1 Dataset Biasing

Since transitioning between the negative and positive $\varphi$ is the slowest process, with the positive $\varphi$ state being less probable, we follow the convention of Klein and Noé [8], Klein et al. [12] and use a

| Experiment | Model Type | Architecture | Parameters |
|---|---|---|---|
| Alanine Dipeptide | Flow Matching | ECNF | 147,599 |
| Alanine Dipeptide | Flow Matching | GVP | 108,933 |
| Alanine Dipeptide | Energy-Based Model | Graphormer | 4,879,949 |
| Dipeptides (2AA) | Flow Matching | ECNF | 1,044,239 |
| Dipeptides (2AA) | Flow Matching | GVP | 735,109 |
| Dipeptides (2AA) | Energy-Based Model | Graphormer | 6,102,153 |

Table 5: Model size of different neural network architectures used in this work.

version of the dataset with bias to achieve nearly equal density in both states, which helps in obtaining a more accurate estimation of free energy. To achieve the biased distribution, weights based on the von Mises distribution, $f_{vM}$, are incorporated and computed along the $\varphi$ dihedral angle as

$$\omega(\varphi) = r \cdot f_{vM}\big(\varphi \mid \mu = 1, \kappa = 10\big) + 1 \tag{31}$$

Where $r$ is the ratio of positive and negative $\varphi$ states in the dataset. To achieve dataset biasing, samples are drawn based on this weighted distribution.

### E.2 Umbrella sampling

Umbrella sampling is a physics-based method used to estimate the free energy profile along a reaction coordinate. It involves selecting a set of points along the coordinate, performing separate simulations around each point using a biasing potential, typically a harmonic restraint, and then combining the resulting data to reconstruct an unbiased free energy landscape. The biasing potential keeps the system near the target point while also promoting sampling of regions that are otherwise rarely visited due to energy barriers.

For alanine dipeptide, Klein et. al[12] ran umbrella sampling simulations with the GFN2-xtb force-field. We utilize the data from the same simulation and treat it as the ground truth value of the free energy projection.

### E.3 Conformer matching

For the generalizability experiments, the bonded graph of a generated sample is inferred using empirical bond distances and atom types in a similar manner to Klein and Noé [8]. The inferred graph is then compared with a reference graph of the molecule and the sample is discarded if the two graphs are not isomorphic.

### E.4 Correcting for chirality

Since SE(3) equivariant neural networks are invariant to mirroring, the Emulator models tend to generate samples from both chiral states. To account for this, we fix chirality *post-hoc* following the convention set by Klein and Noé [8], Klein et al. [12].

### E.5 Time-lagged Independent Component Analysis (TICA)

TICA is a dimensionality reduction technique introduced for analyzing time-series data. In general, it is used to identify directions that maximize the autocorrelation at a chosen lag time. Projecting data onto TICA components yields lower dimensional representations that preserve the system's slowest timescales. Similar to Klein and Noé [8], we construct TICA components using the deeptime library [72] at a lag time of $0.5\ ns$.

### E.6 Model hyperparameters

Each GVP-Boltzmann Emulator model consists of one message-passing GVP layer and one update GVP layer. The alanine dipeptide (ADP) emulators use 5 hidden layers with vector gating, whereas the dipeptide emulators use 9. ADP emulators are configured with 64 scalar features and 16 vector features, while the dipeptide emulators use 128 scalar and 32 vector features.

The Graphormer-based potential models are instantiated with 256-dimensional node embeddings and matching 256-unit feed-forward layers within each transformer block, with a total of 8 layers for ADP and 10 layers for dipeptides. Self-attention is applied with 32 heads over these embeddings, and interatomic distances are encoded using 50 Gaussian basis kernels. The total parameter counts for each model used in this work are reported in Table 5.

## E.7 Endpoint training weights

The Endpoint loss function for training the Boltzmann Emulator is given by:

$$\mathcal{L}_{EP} = \mathbb{E}_{t \sim \mathcal{U}(0,1), (x_1,x_0) \sim C(x_1,x_0)} \left[ \| \frac{\dot{\beta}_t \alpha_t - \beta_t}{\alpha_t} (\hat{x}_1(t, I_t) - x_1) \|^2 \right] \tag{32}$$

Note that, the coefficients $\frac{\dot{\alpha}_t \beta_t - \alpha_t}{\beta_t}$ become divergent near $t \to 1$ as $\beta_1 = 0$. Therefore, in practice, we threshold the min and the max value of these coefficients as follows:

$$t_w = \min(\max(0.005, |\frac{\dot{\beta}_t \alpha_t - \beta_t}{\alpha_t}|), 100) \tag{33}$$

And optimize the following objective:

$$\mathcal{L}_{EPmod} = \mathbb{E}_{t \sim \mathcal{U}(0,1), (x_0,x_1) \sim C(x_0,x_1)} \left[ t_w \| \hat{x}_1(t, I_t) - x_1 \|^2 \right] \tag{34}$$

## E.8 Training protocols

Emulator models for ADP were trained for 1,000 epochs, while those for dipeptides were trained for 12 epochs. Both were trained using the Adam optimizer with a learning rate of 0.001 and a batch size of 512. A learning rate scheduler was employed to reduce the rate by a factor of 2 after 20 consecutive epochs without improvement, down to a minimum of $1e^{-5}$. An Exponential Moving Average (EMA) with $\beta = 0.999$ was applied to the models and updated every 10 iterations. For ADP, batches were coupled using mini-batch optimal transport[73], while for dipeptides independent coupling with rotational alignment was employed. Mini-batch optimal transport was computed using the SciPy `linear_sum_assignment` function [74]. All models were trained on NVIDIA L40 GPUs with a batch size of 512.

The EBMs in both settings are trained with independent coupling. For ADP, the training set consists of 100,000 conformers generated by the emulator, while for dipeptides the training set includes 50,000 conformers generated by the emulator across the 200 dipeptides in the dataset. The ADP EBM is trained for 1,000 epochs, whereas the dipeptide EBM is trained for 10 epochs.

For ADP, the negative time point is sampled from a Gaussian distribution with a standard deviation of 0.025, while for dipeptides it is sampled from a Gaussian with a standard deviation of 0.0125. Both models are optimized using Adam with a learning rate of 0.001. The learning rate is reduced by half after 30 consecutive evaluations/epochs without improvement, down to a minimum of $1e^{-5}$. Training is performed with a batch size of 512.

## E.9 Interpolant Formulation

We specify the interpolant process following the design choices explored in Ma et al. [32]. The Emulator models are trained with linear interpolants while the energy-based models use trigonometric interpolants, both of which satisfy the constraints to generate an unbiased interpolation process.

$$Linear : \alpha_t = 1 - t, \qquad \beta_t = t \tag{35}$$

$$\text{Trigonometric: } \alpha_t = cos(\frac{1}{2}\pi t), \qquad \beta_t = sin(\frac{1}{2}\pi t) \tag{36}$$

Trigonometric interpolants are called general vector preserving interpolants (GVP) in Ma et al. [32]. However, we change the naming of this notation to avoid confusion with geometric vector perceptrons (GVP), which are repeatedly discussed in our paper.

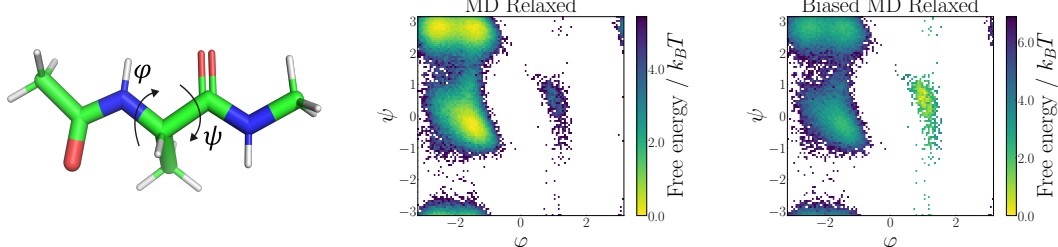

Figure 6: **Alanine Dipeptide System** A visualization of the alanine dipeptide system. Cartoon representation of the alanine dipeptide (left) with its rotatable dihedral angles labeled, Ramachandran plots of unbiased (center) and biased (right) datasets. The biased MD upweights the low frequency mode along the $\varphi$ dihedral.

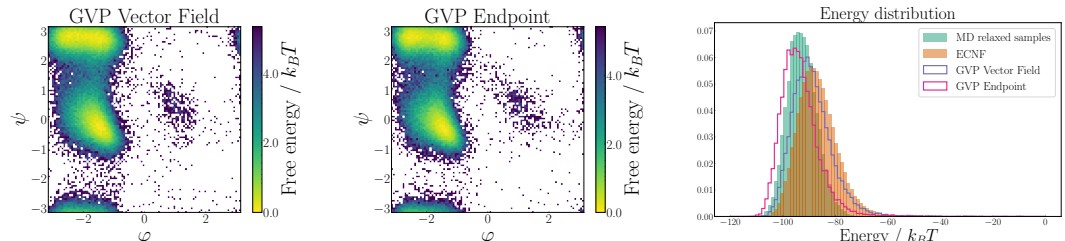

Figure 7: **Evaluation of different emulator models.** The Ramachandran plots of dihedral angles of both endpoint and vector field models are displayed on the left, and center. We see that the energy distributions (right) of both endpoint and vector field emulators as well as a previous method, ECNF, deviate from the distribution of the MD data. However, the Endpoint model distribution deviates the least.

### E.10 Integration scheme

All models were integrated with the adaptive step size DOPRI5 solver implemented in the Torchdiffeq package [75]. The tolerance `atol` and `rtol` values were set to $1e^{-5}$ for alanine dipeptide and $1e^{-4}$ for systems of dipeptides. Vector field model integrals are evaluated from 0 to 1, while endpoint models are evaluated from 0 to $1 - 1e^{-3}$ in order to avoid the numerical instability that occurs with endpoint parametrization at time $t = 1$.

## F   Additional Results

### F.1   Vector Field Vs Endpoint Objectives

Inference results for the Boltzmann Emulators are presented in Table 6 and Figure 7. In this section, we aim to quantify what training objective makes the best emulator and, surprisingly, whether a better emulator will always make a better generator.

| Method | $\mathcal{E}\text{-}W_2$ | $\mathbb{T}\text{-}W_2$ | NLL | NLL std |
|---|---|---|---|---|
| ECNF | $6.22 \pm 0.12$ | $0.27 \pm 0.01$ | $\mathbf{-125.53 \pm 0.10}$ | $\mathbf{5.09 \pm 0.09}$ |
| GVP Vector Field | $4.99 \pm 0.50$ | $0.27 \pm 0.02$ | $-125.42 \pm 0.15$ | $6.92 \pm 0.62$ |
| GVP Endpoint | $\mathbf{3.11 \pm 0.70}$ | $\mathbf{0.26 \pm 0.02}$ | $-92.04 \pm 3.24$ | $175.12 \pm 35.51$ |

Table 6: **Boltzmann Emulator results.** Comparison of NLL and $W_2$ metrics of Boltzmann Emulators across 5 runs ($\pm$ indicates standard deviation). GVP Endpoint emulator captures the energy and torsional target distribution the best. The ECNF model provides the best NLL values despite having the worst $W_2$ metric values, indicating likelihood integration errors for the GVP models. This is also demonstrated with the higher intra-run NLL std deviation values for the GVP models.

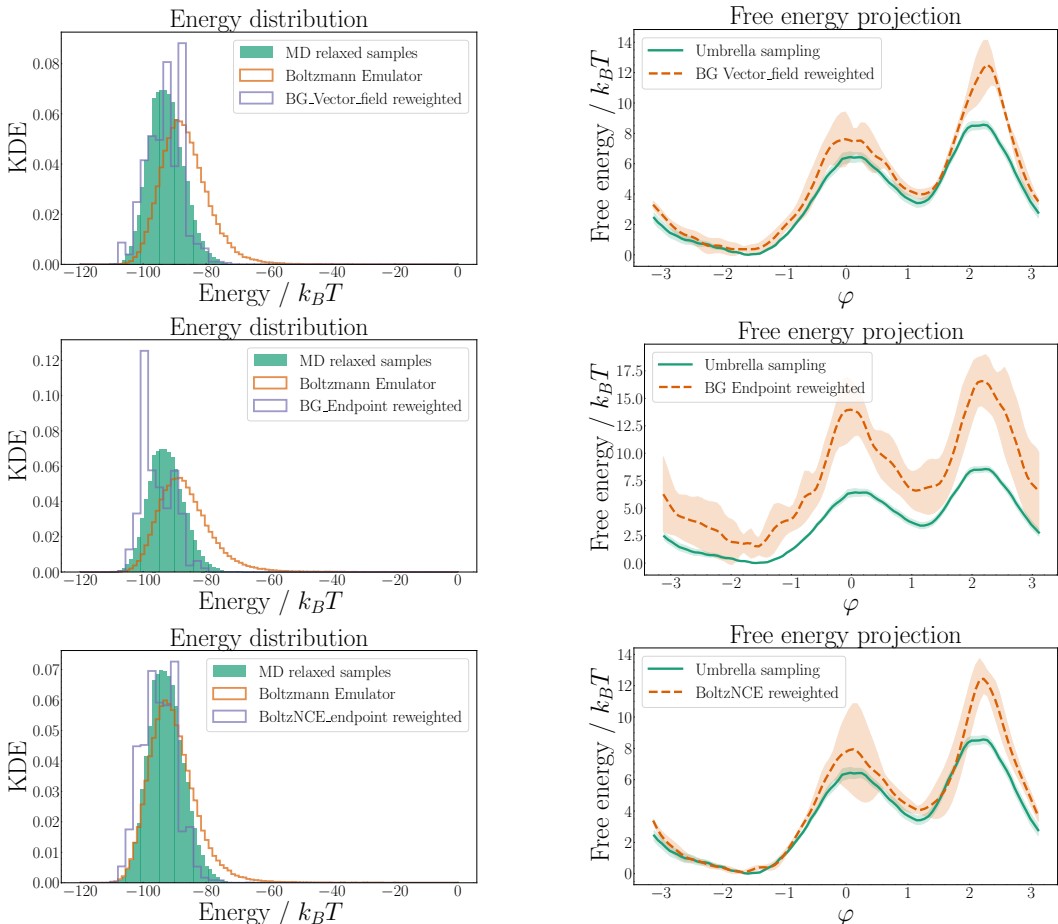

Figure 8: Energy histograms and free energy projections with confidence intervals for the GVP-Vector Field (**top**), GVP-Endpoint (**center**) and BoltzNCE-Endpoint (**bottom**) models.

The energy ($\mathcal{E}\text{-}W_2$) and torsion angle ($\mathbb{T}\text{-}W_2$) Wasserstein-2 distances quantify the discrepancy between the distributions of energies and torsional angles of generated conformers and those in the dataset. The results show that while the $\mathbb{T}\text{-}W_2$ distance remains relatively consistent across all methods, the GVP models capture the dataset's energy distribution better, with the Endpoint model showing the best performance (Figure 7) indicating that it is a very good Boltzmann Emulator on this dataset.

The ECNF and GVP-VF models are comparable on the Negative Log Likelihood (NLL) metric, whereas the GVP-EP model yields the worst values. It is important to note, however, that the endpoint vector field (Eq. 9) diverges at time-point 1. Consequently, the likelihoods for the GVP-EP model were evaluated starting from a later time point $t = 1 - 1e^{-3}$. Furthermore, the divergence at $t \to 1$ can lead to inaccurate likelihood estimates due to instability in the ODE integration. The standard deviation of NLL values within each run is also reported, and the large variance observed for the GVP-EP model further highlights the potential unreliability of its likelihood computations.

### F.2 Boltzmann Generator Results

Energy histograms and free energy projections for GVP Vector Field, GVP Endpoint, and BoltzNCE Endpoint methods are shown in Figure 8. The free energy values and energy histograms match up best with the BoltzNCE Endpoint method.

### F.3 EBM in comparison to the Hutchinson trace estimator

Likelihoods predicted by the EBM are directly compared to the ground truth likelihoods obtained from applying the change of variable equation on the flow matching generated samples. Note

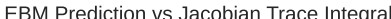

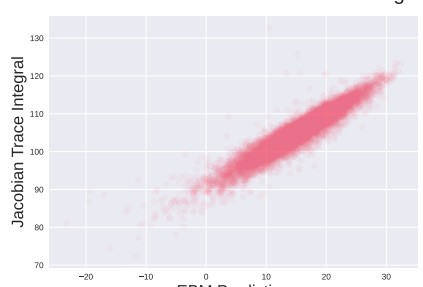 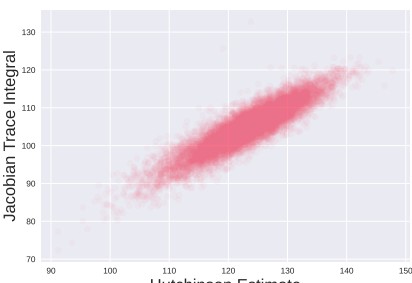

Figure 9: **Likelihood Calculation scatter plots**: log likelihoods estimated by the EBM and Hutchinson estimator vs ground truth estimates from the continuous change of variable equation. EBM predicts likelihoods at high level of accuracy.

| Metric | EBM | Hutchinson 1 call | Hutchinson 2 calls | Hutchinson 4 calls | Hutchinson 8 calls |
|---|---|---|---|---|---|
| Spearman ($\uparrow$) | **0.93** | 0.87 | 0.88 | 0.88 | 0.87 |
| Time ($\downarrow$) | **0.5s** | 8 min | 28 min | 23 min | 23 min |

Table 7: **Likelihood estimation results**: Correlation between likelihood estimation methods and exact likelihoods. EBM performs best.

that the likelihoods estimated by the EBM only estimate the exact likelihoods up to a constant. The Hutchinson trace estimator is also implemented as a comparative benchmark. The Spearman correlation and time required to estimate likelihoods for 10,000 samples is reported in table 7.

The EBM output exhibits strong agreement with the exact likelihoods. Furthermore, the EBM outperforms the Hutchinson estimator, achieving better correlation while being over two orders of magnitude faster at inference on 10,000 samples. Interestingly, increasing the number of estimator calls in the Hutchinson method does not improve its correlation. Due to the use of an adaptive step-size ODE solver, 4 and 8 estimator calls are actually faster than 2 calls.

### F.4 Coupling Function Benchmark

To evaluate the effect of different coupling functions on EBM training, we compare independent coupling to mini-batch OT coupling on ADP. The results are presented in Table 8. The results indicate that both coupling functions can be used to train the EBM and both achieve similar performance.

| Method | $\Delta F$ Error | $\mathcal{E}\text{-}W_2$ | $\mathbb{T}\text{-}W_2$ |
|---|---|---|---|
| Independent Coupling | $0.02 \pm 0.13$ | $0.27 \pm 0.02$ | $0.57 \pm 0.00$ |
| OT Coupling | $0.03 \pm 0.12$ | $0.23 \pm 0.04$ | $0.56 \pm 0.005$ |

Table 8: **Coupling Benchmark.** Coupling functions test to train the EBM model on ADP. Both coupling functions provide similar performance, indicating that the training algorithms are independent of coupling functions.

## G   Dipeptides generalizability results

Quantitative results on the 7 test systems of dipeptides are reported in Table 9. Representative energy histograms and free energy surfaces for the dipeptides are shown in Figure 10. In general, BoltzNCE is able to approximate the right distribution within acceptable error limits at 6x compute time improvement over MD simulations.

| Method | Dipeptide | $\Delta F$ Error | $\mathcal{E}\text{-}W_2$ | $\mathbb{T}\text{-}W_2$ |
|---|---|---|---|---|
| MD | AC | 0.048 | 0.192 | 0.213 |
| TBG-ECNF | AC | 0.009 | 0.202 | 0.290 |
| BoltzNCE | AC | 0.356 | 0.431 | 0.318 |
| MD | ET | 0.069 | 0.152 | 0.182 |
| TBG-ECNF | ET | 0.187 | 0.492 | 0.341 |
| BoltzNCE | ET | 0.222 | 1.329 | 0.280 |
| MD | GN | 0.545 | 0.122 | 0.171 |
| TBG-ECNF | GN | 0.355 | 0.198 | 0.267 |
| BoltzNCE | GN | 0.502 | 1.374 | 0.296 |
| MD | IM | 0.504 | 0.142 | 0.269 |
| TBG-ECNF | IM | 0.026 | 0.430 | 0.362 |
| BoltzNCE | IM | 0.688 | 0.459 | 0.454 |
| MD | KS | 0.044 | 0.138 | 0.232 |
| TBG-ECNF | KS | 0.133 | 0.474 | 0.434 |
| BoltzNCE | KS | 0.477 | 0.419 | 0.626 |
| MD | NY | 0.0003 | 0.198 | 0.229 |
| TBG-ECNF | NY | 0.034 | 0.491 | 0.425 |
| BoltzNCE | NY | 0.090 | 1.293 | 0.591 |
| MD | NF | 0.072 | 0.221 | 0.214 |
| TBG-ECNF | NF | 0.102 | 0.275 | 0.301 |
| BoltzNCE | NF | 0.701 | 2.318 | 0.536 |

Table 9: **Dipeptides results.** Quantitative results of different methods on all 7 dipeptide systems. BoltzNCE delivers acceptable performance while offering a substantial time advantage.

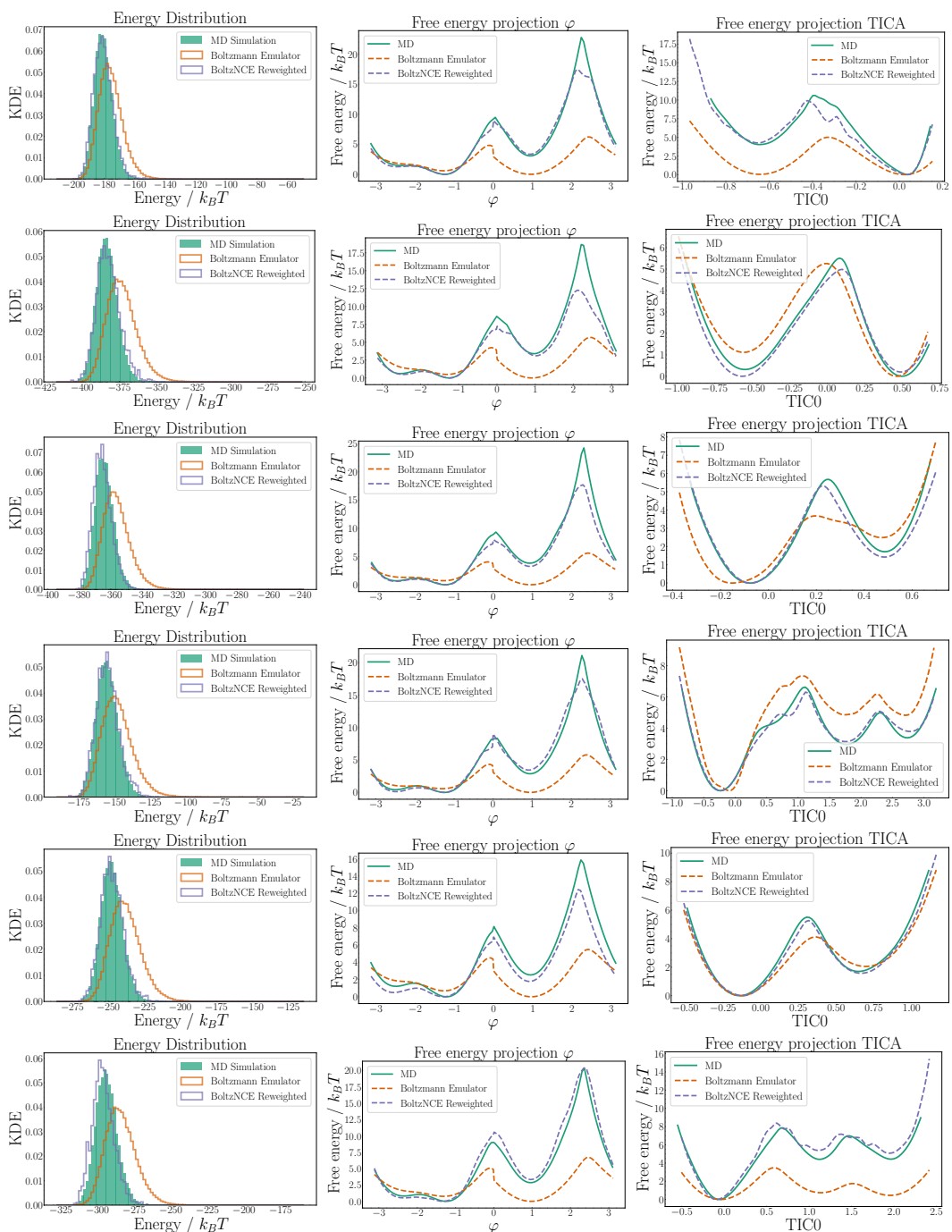

Figure 10: **Qualitative results on dipeptides.** Energy histograms and free energy projections along the $\varphi$ dihedral angle and the first TICA component for test dipeptides. In order from top to bottom, the figures represent results on the following dipeptides: AC, ET, GN, IM, KS, NF. In all cases, BoltzNCE achieves good approximations of the energy distribution and free energy surfaces.

