# OpenReview forum: "BoltzNCE: Learning likelihoods for Boltzmann Generation with Stochastic Interpolants and Noise Contrastive Estimation"
_NeurIPS.cc/2025/Conference — NeurIPS 2025 poster_

### Official Review · Reviewer_UPgQ · 2025-06-01

**Clarity:** 3
**Significance:** 4
**Originality:** 3
**Rating:** 5
**Confidence:** 4

**Summary:**

The paper introduces BoltzNCE, a way to train an EBM and continuous normalizing flow simultaneously. One caveat of cnfs is that although they can be trained simulation free tend to have unstable/inefficient likelihood computations, making them infeasible for reweighting. The authors propose to combine a FM loss together with training a family of time-parameterized EBMs that should approximate the distribution of the cnfs. Luckily, this can be done using score matching and Info-NCE, which both do not require solving an ode. The approach is then compared for free energy calculations on alanine dipeptide.

**Questions:**

1) The INFO NCE loss is not clear: while I am family with the idea of NCE losses in EBMs, i do not understand why that is equivalent to the NLL (is that somehow like 1d integration??). please add some more explanation on that. why is it reasonable to sample negative points around t with a small variance?

2) In the contributions you write "We show that the learned likelihoods can be used to reweight conformations to match the Boltzmann distribution. To the best of our knowledge, this is the first method to recover the Boltzmann distribution without requiring exact likelihood computations." I do not understand this, there have been a lot of papers recovering Boltzmann dists using approximate likelihoods and CNFs?

3) I am not super familiar with molecular simulations, so please explain for me (not necessarily in the paper), why the free energy of a certain angle is such a good proxy for likelihood evals.

4) Does training the EBM on the side deteriorate the sampling (i.e, W_2) performance of the FM method?

**Ethical Concerns:**

["NO or VERY MINOR ethics concerns only"]

**Final Justification:**

My questions are answered, this paper deserves an acceptance imo.

**Limitations:**

yes

**Quality:**

3

**Strengths And Weaknesses:**

Stengths:

1) The approach is to the best of my knowledge new. While many papers have tried training EBMs  on their own, guiding EBMs by a CNF seems smart.

2) The paper is also (up to a few problems that can be misunderstandings on my end) very clearly written, and the approach simple to understand.

3) The motivation is clear and the experiments are encouraging.

4) The ablations on why both the score and NCE loss are needed are very neat.

Weaknesses:

1) I would like to see experimental evaluation on a simple imaging dataset. One can easily train a FM model on MNIST, and then see if one can reuse the EBM from your approach as a regularizer (i.e. prior) in a simple inverse problem such as deblurring or inpainting. This would greatly strengthen the paper, as the usefulness would span to a different subfield and show that the method can also work there. Given that this is a method/experimental paper, I would expect more comparisons.

2) I had a lot of trouble understanding some bits of the paper: more precisely the INFO NCE loss needs more clarity.

3) As a baseline one could also consider distilling the FM model, i.e., train a model which has similar likelihoods/odes but with much fewer steps. This would essentially serve the same purpose as your EBM. A discussion of this (if you do not consider it a strong baseline) or a small experiment would also strengthen the paper.

4) While the paper makes a very good experimental point, that both the NCE and the SM loss are needed, I would also like some (heuristic) explanation why that is.

5) Citations are unfortunately very sparse in the text. While every paper used is cited in the intro/related work, I would also like to see that when introducing the methods so it is very clear where the method is from and where one can read it up. Otherwise, one could confuse some of those as your contributions.

---

> ### Author Rebuttal · Authors · 2025-07-31
>
> We thank the reviewer for their time and their helpful comments. We are glad that the reviewer found our paper novel and well written. We would also like to mention that in addition to results on alanine dipeptide, we now include new experiments demonstrating transferability For transferability we train on a set of dipeptides and infer on dipeptide systems not present in the training set. With a bit of fine-tuning of the EBM on the unseen dipeptide, we obtain accurate free energy estimates and low energy/torus W-2 scores. Visually our energy histograms have nearly perfect overlap and free energy projections have great agreement with the test set.
>
> **Imaging and other experiments:**
>
> >I would like to see experimental evaluation on a simple imaging dataset.
>
> We thank the reviewer for their suggestion on solving the inverse problems on images. We definitely think this could be an interesting direction of research and experiment/algorithm design with an EBM deserves further thought. However, for the rebuttal period, we had to focus on other requested experiments on molecular systems (transferability and scalability) as efficiently solving the Boltzmann distribution is the primary purpose of this work. Further details on this can be seen in our reply to reviewer 3 vcq2.
>
> **InfoNCE objective:**
>
> >The INFO NCE loss is not clear...
>
> Thank you for your comments and questions, we have improved the writeup for the InfoNCE objective in our manuscript. The InfoNCE objective is an orthogonal objective function that maximizes the conditional likelihood $\rho_\theta(t|x_t)$.
>
> $$\rho_\theta(t|x) = \frac{\rho_\theta(t,x_t)}{\rho_\theta(x_t)} = \frac{\rho_\theta(t,x_t)}{\int\rho_\theta(t',x_t)dt'} = \frac{\exp(E_\theta(t,x_t))}{\int \exp(E_\theta(t',x_t))dt'}$$
>
> where $E_\theta$ is the energy based model. This leads us to minimize the negative log likelihood of the conditional:
>
> $$\mathcal{L_{\text{NLL}}}(\theta) = E_{x_t}\left[-\log \frac{\exp(E_\theta(t,x_t))}{\int \exp(E_\theta(t',x_t))dt'}\right]$$
>
> The intractable integral in the denominator is approximated by appropriately sampling a set of *negative* time points $t'$ resulting in the InfoNCE loss:
>
> $$\mathcal{L_{\text{InfoNCE}}}(\theta) = E_{x_t}\left[ -\log \frac{\exp(E_\theta(t,x_t))} {\sum_{t' \in \{\tilde{t}\} \cup t } {\exp(E_\theta(t',x_t))}}\right]$$
>
> >please add some more explanation on that. why is it reasonable to sample negative points around t with a small variance?
>
> In practice, on the interpolant process, only time points $t'$ close to $t$ will have a non-negligible likelihood $\rho_\theta(t', x_t)$, since $t$ also determines the amount of noise added to the clean sample. Therefore, it is reasonable to sample $t'$ using a distribution with small variance centered around $t$.
>
> > While the paper makes a very good experimental point, that both the NCE and the SM loss are needed, I would also like some (heuristic) explanation why that is.
>
> Thank you for your question, we expand on it further below:
>
> The score matching loss minimizes the Fisher divergence between the model and data distributions, and therefore also implicitly minimizes the KL divergence. However, as demonstrated in our 2D experiments, while the score matching objective effectively identifies the modes of the data distribution, it does not reliably capture their relative weights. The InfoNCE loss, being orthogonal to score matching, serves as a complementary regularizer that helps the model learn more accurate relative weighting between modes.
>
> **Distillation of FM model as a baseline**
>
> > As a baseline one could also consider distilling the FM model, i.e., train a model which has similar likelihoods/odes but with much fewer steps
>
> Thank you for the question. Most FM distillation methods we are familiar with fall under the category of consistency models. Due to their algorithmic design, these methods do not support likelihood estimation. The only method we are aware of that potentially reduces the number of integration steps is rectified flows, however, even this approach does not address the challenge of scaling inference with dimensionality. Furthermore, since our FM models are already trained with mini batch OT, rectified flow distillation may not reduce the transport cost by much.
>
> **Citations**
>
> > I would also like to see ['citations'] when introducing the methods
>
> Thank you for your comment. We have now added citations throughout the Methods and Results sections, including references related to stochastic interpolants, vector field and score matching objectives, endpoint parameterization, optimal transport, GVP, Graphormer, and more. Additionally, we have included more citations to several seminal works on score matching and flow matching in the main text.
>
> **Free Energy**
>
> >please explain for me (not necessarily in the paper), why the free energy of a certain angle is such a good proxy for likelihood evals.
>
> We believe the free energy projection of a system is a relevant baseline for the following two reasons:
>
>
> * **It represents a high dimensional integral of probability**: The equation of a free energy projection along a reaction coordinate is given by:
>
>   $$
>   F(r') = -K_BT \ln \rho(r')
>   $$
>
>   where $\rho(r')$ is the probability density of observing the system at position $r'$ along the coordinate,
>
>   $$
>   \rho(r') = \frac{1}{Z} \int_x \delta(r(x) - r') e^{(-U(x)/K_BT)} \, dx
>   $$
>
>   In principle, if we solve the above integral along all points of the reaction coordinate, we solve a ($D-1$) dimensional integral, where $x \in R^D$. Thus, this serves as a good metric on how well the model matches the ground truth $R^D$ distribtuion.
>
>   For dipeptides the process of going from the negative $\varphi$ angle to the positive $\varphi$ angle is considered a slow process as there are regions of high energy/low probability in between the two. Therefore, this serves as an ideal reaction coordinate to study dipeptide systems.
>
> * **Domain Relevance:**  Applied studies in the biophysical/ biochemical/ structural biology domain work on elucidating the free energy projection along a *reaction coordinate*. This helps the researchers identify the relative stability of different modes along the coordinate as well as the rate of reaction along the coordinate.
>
> **Other Questions**
>
> > In the contributions you write "We show that the learned likelihoods can be used to reweight conformations to match the Boltzmann distribution. To the best of our knowledge, this is the first method to recover the Boltzmann distribution without requiring exact likelihood computations." I do not understand this, there have been a lot of papers recovering Boltzmann dists using approximate likelihoods and CNFs?
>
> Thank you for your question. We do want to emphasize here that, at the time of submission, we were only aware of Boltzmann generator methods that compute likelihoods using the change of variable equation (both discrete and continuous versions). For example, CNFs use the continuous change of variable equation to compute likelihoods. We call these *exact likelihoods* methods. As of now, we are only aware of only one other (concurrent) work [1] that does not use exact likelihoods.  We will revise our claim accordingly and clarify that our method also demonstrates strong performance in transferabile settings.
>
> > Does training the EBM on the side deteriorate the sampling (i.e, W_2) performance of the FM method?
>
> Thank you for your question, the EBM does not interfere with the training of the FM model and is only trained to match the likelihoods of samples generated by the FM model. Therefore training it on the side does not affect the performance of the FM model.
>
>
> [1] T. Akhound-Sadegh et al., “Progressive Inference-Time Annealing of Diffusion Models for Sampling from Boltzmann Densities,” arXiv.org, 2025. https://arxiv.org/abs/2506.16471 (accessed Jul. 30, 2025).

---

> > ### Comment · Reviewer_UPgQ · 2025-08-01
> > **rebuttal**
> >
> > I thank the authors for their rebuttal. I think it is a neat work and deserves an acceptance. All my questions are answered and i am raising my score to 5. Still if this approach could yield good priors for inverse problems would be a super nice application, especially if this could help to bridge the gap of using flow models as regularizers without backprop.

---

> > > ### Author Response · Authors · 2025-08-06
> > >
> > > We thank the reviewer and are pleased that the reviewer is satisfied with out response. We agree with the reviewer that our method may be impactful for inverse problems and should be explored more in future works.

---

### Official Review · Reviewer_vcq2 · 2025-06-12

**Clarity:** 3
**Significance:** 2
**Originality:** 2
**Rating:** 4
**Confidence:** 4

**Summary:**

The paper introduces BoltzNCE, a scalable and simulation-free method for training energy-based models (EBMs) to estimate likelihoods for reweighting samples from a generative model toward a target distribution. The approach combines noise contrastive estimation and score matching within the stochastic interpolant framework, avoiding the expensive Jacobian trace computations required by normalizing flows. The learned EBM density enables accurate reweighting of generated samples to recover the Boltzmann distribution. The method is validated on toy datasets and the alanine dipeptide molecule, achieving performance comparable to exact-likelihood normalizing flows while offering orders-of-magnitude faster inference.

**Questions:**

1. Echoing the above concerns, how does the EBM perform when across the metrics when reweighting samples for alanine dipeptide and for larger systems such as the tetrapeptide and hexapeptide variants of alanine (AL4 & AL6), as well as for free energy difference? And, how does the EBM perform when transferring across different systems?

2. It is not immediately clear to me how the statement made on line 54 relates to the density chasm problem. The statement on line 54 is “We tackle this issue by annealing between a simple noise distribution and the data distribution using stochastic interpolants” where the issue here is the density chasm problem mentioned in the previous paragraph. In this case, NCE distinguishes between a sample drawn from the interpolation and a negative sample drawn from a narrow Gaussian centered at that sample. Would the authors please elaborate more on this?

3. Algorithm 1 in appendix B states the EBM uses data samples but the method states the samples at t=0 comes from the generative model. Can the authors clarify this?

4. From what I understand, the framework currently operates with a generative model and EBM being trained separately. I am curious to see how does the generative model perform if the score and vector field are parameterized using \nabla E_\theta from the EBM?

Should the authors address my current concerns (specifically, the additional experiments with scalability to AL4 / AL6 and transferability mentioned in the first point), I would consider raising my score.


Not questions but typos found:
- line 146: enery -> energy
- appendix eq 21: the dots should be \cdot for t \cdot x_1 + (1 - t) \cdot x_0

**Ethical Concerns:**

["NO or VERY MINOR ethics concerns only"]

**Final Justification:**

The authors have addressed my questions and concerns in the rebuttal, and I appreciate the additional experiments provided.

I will raise my score to a 4, as it better reflects the paper’s strengthened experimental design and results. While my concerns have been addressed, the work does not introduce a fundamentally novel methodology or conceptual insight. As such, I do not find grounds for awarding a 5, but I still recommend acceptance.

**Limitations:**

Yes. No potential negative societal impact is involved.

**Paper Formatting Concerns:**

No concerns.

**Quality:**

2

**Strengths And Weaknesses:**

**Strengths**
1. The paper is clearly written and motivated.
2. The proposed method (Vector field) achieves comparable performance to normalizing flows on free energy difference estimation while being significantly faster, encouraging the possibility of scaling to larger systems.
3. The authors provide a simulation-free method to train EBMs through the stochastic interpolant framework.

**Weaknesses**
1. While the method demonstrates accurate likelihood estimation on alanine dipeptide, evaluating it on larger systems - such as tetrapeptides or hexapeptides - would better showcase its scalability and inference efficiency. Additionally, for alanine dipeptide, in table 1, reporting energy W2, torus W2, etc for both the proposal and the reweighted samples would help support the effectiveness of the EBM.
2. Although the GVP-based models support the ability to transfer across systems, the authors do not explore transferability of the EBM across systems. Demonstrating the method’s ability to transfer across molecular systems would strengthen its practicality.

---

> ### Author Rebuttal · Authors · 2025-07-31
>
> We thank reviewer vcq2 for their insights and are glad that they find our work well motivated. We are happy to see that the reviewer agrees with us that our EBM method achieves performance comparable to exact-likelihood normalizing flows while offering orders-of-magnitude faster inference.
>
>
> **2. Additional Experiments**
>
> As suggested by the reviewers, we conducted additional experiments to evaluate the transferability of our method. Specifically, we train our model on 200 dipeptides and test it on 7 unseen dipeptides, closely following the experimental setup described in [1]. Similar to our ADP experiments, we bias the training set to ensure adequate coverage of all relevant modes along the $\varphi$ angle.
>
> To establish a molecular dynamics baseline, we run 1 μs simulations and compute E-W2, T-W2, and $\Delta F$ estimation errors with respect to the test set provided in [1]. We also train the model from [1] on our biased dataset. However, inference with this model requires several days of computation, so we were unable to obtain results for it within the rebuttal period. We will include these results in the camera-ready version.
>
> During inference, we fine-tune our pre-trained EBM using samples generated for each unseen dipeptide. Additionally, we exclude the top 0.2% of importance weights during reweighting, following the approach introduced in [2], which significantly improves the stability of the reweighting process. Our results show that the method successfully transfers across systems while offering a substantial inference-time advantage over MD and ECNF baselines. The performance of our method compared to the MD baseline across the 7 dipeptides is summarized in the table below.
>
>
> | Metric              | Energies W2   | Angles W2    | $\varphi$ $\Delta F$ Error  |Inference time|
> |:--------------------|--------------:|-------------:|-------------------:|-------------------:|
> |  MD Baseline | 0.17 ± 0.03   | 0.22 ± 0.03  | 0.18 ± 0.22 | 1 day |       |
> |  BoltzNCE | 0.94 ± 0.49   | 0.54 ± 0.12  | 0.56 ± 0.33 | 3.7 hours |       |
>
>
> As shown in the table above, our method achieves a 6× speedup over conventional molecular dynamics while producing W2 and $\Delta F$ metrics that closely match the ground truth. Visually, the reweighted energy histograms show near-perfect overlap, and the predicted free energy projections align closely with the ground truth. These results support the conclusion that our method generalizes well and is transferable for inference beyond the training data.
>
> **Reweighting Improves Results**
> > "reporting energy W2, torus W2, etc for both the proposal and the reweighted samples would help support the effectiveness of the EBM"
>
> This is a very interesting idea and provides a great opportunity to show case the strength of our EBM model. We report energy W2 and torus W2 for both the proposal and reweighted samples in the table below. In all cases, we see that reweighting significantly improves generated distributions. The EBM (BoltzNCE Vector field) is able to efficiently reweight samples from the GVP vector field model and beats the ECNF baseline on these metrics
>
>
> | File                          | Proposal E-W2 | Reweighted E-W2 | Proposal T-W2 | Reweighted T-W2 |
> |:-----------------------------|-----------------------------:|---------------------------------:|------------------------------:|---------------------------------:|
> | ECNF        | 8.08±0.56                    | 0.37±0.02                        | 1.10±0.01                     | 0.59±0.00                        |
> |GVP Endpoint           | **6.19±1.08**                    | 2.88±0.01                        | 1.12±0.01                     | 0.58±0.01                        |
> |GVP Vector field                 | 7.20±0.13                    | 0.46±0.05                        | **1.09±0.01**                   | 0.60±0.00                        |
> |BoltzNCE Endpoint        | 6.24±0.52                    | 2.78±0.04                        | 1.12±0.01                     | 0.59±0.01                        |
> | BoltzNCE Vector field         | 7.12±0.15                    | **0.27±0.02**                        | 1.12±0.00                     | **0.57±0.00**                        |
>
> **3. Questions**
> > It is not immediately clear to me how the statement made on line 54 relates to the density chasm problem.
>
> Thank you for the question. The density chasm problem arises when the samples drawn from the 2 distributions are easily distinguishable. Such a case results in vanishing gradients[3] under the NCE objective. Therefore to narrow the gap between the distributions that are compared in the contrastive loss we employ annealing through the use of stohcastic interpolant and ensure that the negative samples are in closely proximity to the positive sample.
>
> > Algorithm 1 in appendix B states the EBM uses data samples but the method states the samples at t=0 comes from the generative model.
>
> Good catch! We have fixed it to reflect that we train the EBM on generated samples.
>
> >I am curious to see how does the generative model perform if the score and vector field are parameterized using $\nabla E_\theta$ from the EBM?
>
> This is an interesting point that is worth exploring further. Our method takes advantage of a good sampler (Flow Matching) and a good likelihood estimator (EBM). We leverage both these properties to get the best of both worlds. For the scope of this rebuttal, however, we mainly focused on experiments related to transferability and leave sampling with $\nabla E_\theta$ to future work.
>
>
> In conclusion, we thank the reviewer for their great suggestions of extra experiments. We believe that the experiments they proposed have truly strengthend our paper and we hope that they see our novel experiments on transferability as even stronger proof of our method. We also show that our EBM is well correlated with likelihoods from the Jacobian trace integral with orders of magnitude speed up. With this, we hope the reviewer considers improving their score on our paper.
>
> [1]Leon Klein and Frank Noé. Transferable boltzmann generators. arXiv preprint arXiv:2406.14426, 2024.
>
> [2]Tan, Charlie B., Avishek Joey Bose, Chen Lin, Leon Klein, Michael M. Bronstein, and Alexander Tong. "Scalable equilibrium sampling with sequential boltzmann generators." arXiv preprint arXiv:2502.18462 (2025).
>
> [3]Liu, B., Rosenfeld, E., Ravikumar, P., & Risteski, A. (2021). Analyzing and improving the optimization landscape of noise-contrastive estimation. arXiv preprint arXiv:2110.11271.

---

> > ### Comment · Reviewer_vcq2 · 2025-08-02
> >
> > Thank you for your detailed response to my concerns and questions. Most of the issues I raised regarding the additional experiments have been addressed such as the transferability and reweighting experiments. While this work would further benefit from scaling to larger systems (such as tetrapeptides and hexapeptides), I ultimately find the revised version stronger and more compelling. I will raise my score and recommend the paper for acceptance.

---

> > > ### Author Response · Authors · 2025-08-06
> > >
> > > We thank the reviewer for their comments and appreciate the score increase. We agree that scaling the method to larger systems is an important task for the future

---

### Official Review · Reviewer_WnHv · 2025-07-02

**Clarity:** 2
**Significance:** 4
**Originality:** 3
**Rating:** 5
**Confidence:** 4

**Summary:**

The paper addresses the accuracy of learned generative models for Boltzmann distributions (Boltzmann generators of short). When these models are not perfect, the estimation of downstream quantities of interest is improved by weighting generated samples with the ratio of true and estimated probabilities. However, the calculation of the estimated probability is expensive for the currently best models (continuous normalizing flows). Therefore, the authors propose to train an additional energy-based model to calculate these probabilities more efficiently. The key innovation is a simulation-free and thus fast training algorithm for the additional network.

**Questions:**

* Can you add proper derivations of key formula?
* Can you add a table saying exactly which algorithms are used in which experiment?
* Can you provide more extensive experiments, or at least demonstrate the agreement between the EBM and the generative model?

**Ethical Concerns:**

["NO or VERY MINOR ethics concerns only"]

**Final Justification:**

I thank the authors for addressing my concerns and would like to encourage them to implement these improvements (especially the ones concerning the theory) as far as possible in the final version. Moving a non-essential section to the appendix to make room for more detailed explanations and derivations of the essential parts is a good idea. I've raised my score to "Accept".

**Limitations:**

The authors acknowledge that experiments restricted to the alanine dipeptide have limited significance. They also mention the limited accuracy of the probability estimates from the EMB, although it is unclear why this has not been investigated in more detail (the item 2. under "Weaknesses").

**Paper Formatting Concerns:**

none.

**Quality:**

2

**Strengths And Weaknesses:**

The proposed idea of simulation-free training for an auxiliary energy-based model for efficient reweighting is very good (although the idea was in the air, given the field's recent evolution). The experiments demonstrate that the new method achieves accuracy competitive with a continuous normalizing flow at 1/50 ... 1/100 of the inference time, which is highly practical relevant.

However, the paper has several major shortcomings:

1. Although there is a lot of theory, the presentation completely lacks derivations of the formulas, nor have sections 3 and 4 any citations. Thus, it is next to impossible to check if the results are correct. In particular, I have concerns with the coupling $C(x_0, x_1)$ between data and latent samples, which must be a joint distribution whose marginals $p(x_0) = \int C(x_0, x_1) dx_1$ and $q(x_1)=\int C(x_0, x_1) dx_0$ are the data and latent distribution respectively. While it is known that the objectives eq. (9), (10) and (16) converge for any valid coupling, this is probably not true for the InfoNCE objective (15). I think that the coupling here must be the optimal coupling resulting from Schrödinger bridge matching, as derived in Shi et al. "Diffusion Schrödinger bridge matching", NeurIPS 2023. The authors mention briefly (lines 183-184) that they approximate the coupling by minibatch optimal transport via the Hungarian algorithm, but they fail to discuss to what degree this approximation fulfills the requirements for the coupling, nor what these requirements are in the first place.

2. In the experiments, it is hard to understand exactly which sub-algorithms according to which formulas are included or excluded in the different model variants ("GVP Vector Field", "GVP Endpoint", "BoltzNCE reweighted", "BoltzNCE Vector Field", and "BoltzNCE Endpoint") mentioned in the diagrams and tables. The information can probably be deduced somehow from the text, but should be made obvious in a table or similar to allow for unambiguous replication of the experiments. Moreover, the "Umbrella Sampling" method, which apparently serves as a ground truth estimate, is entirely undefined. It is also unclear why reweighting was not included in the experiment in figure 4 and table 1 and how other methods fail to estimate the free energy in figure 5.

3. The experiments are somewhat weak, as they only cover the alanine dipeptide (although table 1 does not even say what the data are). For a NeurIPS publication, I would expect a more extensive set of experiments. Moreover, when samples are generated with one model, but weighted by another, it must be ensured that both models agree on the generative probability. Since the probabilities of the generative model can in principle be calculated via the continuous change-of-variables formula (even if this is too expensive for most practical applications), this comparison can and should be conducted.

---

> ### Author Rebuttal · Authors · 2025-07-31
>
> We thank the reviewer for their feedback and we are happy to see they agree that our method provides orders of magnitude speed up with competitive accuracy.
>
> **1. Experimental Results**
>
>
> > The experiments are somewhat weak, as they only cover the alanine dipeptide
>
>
> We agree that additional experimental validation would benefit our paper and provide stronger proof that our method works. To this order, we show that our method is **transferable** by training on 200 dipeptides and testing on 7 unseen dipeptides. With a bit of fine tuning on the unseen dipeptide, we achieve good performance while attaining a significant time advantage over MD and benchmark ECNF methods. Visually our energy histograms have nearly perfect overlap and free energy projections have great agreement with the test set. For more details on transferability, refer to our reply to reviewer vcq2. These results are discussed in depth in our reply to reviewer vcq2 (reviewer 3).
>
> >when samples are generated with one model, but weighted by another, it must be ensured that both models agree on the generative probability
>
> We agree! To showcase this, we calculated Spearman correlation coefficients between the EBM output and exact likelihood as calculated by the Jacobian trace integral for the alanine dipeptide syteme. Our final paper version will include the scatterplots but we report just the quantative value below.
>
> As shown in Table 1, the EBM output exhibits strong agreement with the flow-matching likelihoods. Moreover, we found that our EBM outperforms the Hutchinson estimator, achieving better correlation while being over two orders of magnitude faster at inference on 10,000 samples. Interestingly, increasing the number of estimator calls in the Hutchinson method does not improve its correlation. Due to the use of an adaptive step-size ODE solver, 4 and 8 estimator calls are actually faster than 2 calls.
>
> | Metric     | EBM      | Hutchinson (1 estimator call) | Hutchinson (2 estimator calls) | Hutchinson (4 estimator calls) | Hutchinson (8 estimator calls) |
> |:-----------|:--------:|:-----------------------------:|:-----------------------------:|:-----------------------------:|:-----------------------------:|
> | Spearman R | **0.93** | 0.87                          | 0.88                          | 0.88                          | 0.87                          |
> | Time       | **0.5 s**| 8 min                         | 28 min                        | 23 min                        | 23 min                        |
>
> **2. Formulas and equations**
> >lacks derivations of the formulas, nor have sections 3 and 4 any citations
>
> Thank you for the comments. We have now included multiple new citations throughout the methods and results sections so that the reader can follow where some parts of the method are from. Some of the main papers we cite heavily include:
>
> * For stochastic interpolant equations
>     * Nanye Ma, Mark Goldstein, Michael S Albergo, Nicholas M Boffi, Eric Vanden-Eijnden, and Saining Xie. Sit: Exploring flow and diffusion-based generative models with scalable interpolant transformers. In European Conference on Computer Vision, pages 23–40. Springer, 2024.368
>
> * For Boltzmann Generator formulations and equations
>     * Leon Klein, Andreas Krämer, and Frank Noé. Equivariant flow matching. Advances in Neural  Information Processing Systems, 36:59886–59910, 2023
>
> * For Endpoint Objective
>     * Bowen Jing, Bonnie Berger, and Tommi Jaakkola. Alphafold meets flow matching for generating protein ensembles. arXiv preprint arXiv:2402.04845, 2024
> * For mini-batch OT
>     * Alexander Tong, Kilian Fatras, Nikolay Malkin, Guillaume Huguet, Yanlei Zhang, Jarrid Rector-Brooks, Guy Wolf, and Yoshua Bengio. Improving and generalizing flow-based generative models with minibatch optimal transport. arXiv preprint arXiv:2302.00482, 2023.459
>
> We have also included citations to other seminal score matching and flow matching works.
>
> **3. Coupling concerns**
> > The coupling here[in the EBM] must be the optimal coupling resulting from Schrödinger bridge matching
>
> Thank you for your comment. To test your hypothesis, we trained versions of the EBM model with both OT and independant coupling. We found that the models converged in both cases and we saw only minor changes in model performance. We report the performance of both methods in the table below. With this we conclude that OT coupling may not be necessary for model convergence.
>
>
>
> |   | $\Delta F$ Error | E W-2 | T W-2 |
> |----------|----------|----------|----------|
> |OT Coupling          |     0.03   ±0.12      |  0.23±0.04       |      0.56±0.005    |
> |Independent Coupling|     0.02 ±0.13   |   0.27±0.02       |       0.57±0.00   |
>
> > The authors mention briefly (lines 183-184) that they approximate the coupling by minibatch optimal transport via the Hungarian algorithm, but they fail to discuss to what degree this approximation fulfills the requirements for the coupling, nor what these requirements are in the first place.
>
> Thank you for your comment. As shown in [1], minibatch OT approximates entropic regularize OT coupling of the two distributions. The hungarian algorithm is of similar flavor and is mainly used to for scalability to larger batch sizes.
>
> **4. Generator and Emulator Clarifications**
> >  it is hard to understand exactly which sub-algorithms according to which formulas are included or excluded in the different model variants
>
> We apologize for the confusion, we added the following descriptions more explicitly in the tables and figures in our newest version in a similar manner as the table below.
>
> | Method                            | FM Objective | Likelihood Estimation |
> | --------------------------------- | ------------ | --------------------- |
> | GVP Vector field                  | Vector Field | Jac-trace integral    |
> | GVP Endpoint                      | Endpoint     | Jac-trace integral    |
> | BoltzNCE Vector field             | Vector Field | EBM forward pass      |
> | BoltzNCE Endpoint                 | Endpoint     | EBM forward pass      |
>
> We have now also added a description of the umbrella sampling method in our paper.
>
> > It is also unclear why reweighting was not included in the experiment in figure 4 and table 1
>
> Figure 4 and Table 1 aimed to compare the performance of the *emulators* not the *generators*. ie. we wanted to show the impact of the choice of objective functions without the influence of reweighting. Furthermore, our results showed that a better emulator does not neccesarily create a better generator due of the design constraints that generators have to follow. However, we agree that this section may confuse readers and detract from the main point of the paper and have moved this discussion to the appendix. We have, however, now added additional W-2 results, before and after reweighting on the Boltzmann Generators comparison table. See further detail on this in our response to reviewer vcq2.
>
> > How other methods fail to estimate the free energy in figure 5
>
> Thank you for the question. We have provided free energy projections for the other methods in the supplementary material. We hypothesize that the other methods have slightly more inaccurate free energy projection predictions due to inaccurate likelihood estimates that can occur due to error propogation during ODE integration. This is especially prevelant when one tries to integrate likelihoods on CNF models trained with the endpoint objective.
>
> Overall we thank the reviewer for their valuable comments, we hope that they see our novel experiments on transferability provides even stronger proof that our method unlocks the potential for a truely boltzmann generator. We also address concerns regarding couplings for the EBM with emperical results and provide additional clarification on methods and equations. With this, we hope the reviewer considers improving their score on our paper.
>
>
> [1] Tong, Alexander, et al. "Simulation-free schrodinger bridges via score and flow matching." arXiv preprint arXiv:2307.03672 (2023).

---

### Official Review · Reviewer_jMkc · 2025-07-03

**Clarity:** 3
**Significance:** 2
**Originality:** 3
**Rating:** 4
**Confidence:** 4

**Summary:**

This paper proposes a scalable, simulation-free framework for learning sample likelihoods from energy-based models to accurately model the Boltzmann distribution. By combining score matching and noise contrastive estimation with stochastic interpolants, the method avoids the costly exact likelihood computations that are usually impractical for large molecular systems. This approach enables efficient estimation of free energy differences and recovery of the Boltzmann distribution, delivering comparable accuracy with substantial computational speedup.

**Questions:**

- Is it possible to just use EBM for sampling, since it already provide a score of the model density.
- In your dipeptide experiments, how does the model perform if the emulator training dataset are more biased?

**Ethical Concerns:**

["NO or VERY MINOR ethics concerns only"]

**Final Justification:**

The additional experiment with alternative architectures has address my concerns that the performance gain is from methods itsefl

**Limitations:**

yes

**Quality:**

3

**Strengths And Weaknesses:**

**Strengths**
The authors combine stochastic interpolants with the Boltzmann Generator and propose a contrastive learning approach to train the energy-based model (EBM), which circumvents the need to compute the intractable integral in the normalizing constant. The method is evaluated on both 2D toy examples and dipeptide data, showing performance improvements over ECNF.

**Weakness**
- The method requires training an additional EBM, whereas other approaches typically train only a single CNF model.
- The architecture choices are complex, yet architectural ablations are missing. A discussion is needed to clarify that performance gains are not merely due to increased parameter counts.
- The endpoint models diverge near t = 0, requiring manual truncation of the integration. This instability reduces the effectiveness of likelihood-based sampling.
- If the Boltzmann  emulator  is trained on biased data with missing modes, the EBM is unable to recover those missing modes.

---

> ### Author Rebuttal · Authors · 2025-07-31
>
> We thank the reviewer for their thoughtful review and questions, and we are pleased that they found our approach practical for scaling to larger systems. We would also like to mention that in addition to results on alanine dipeptide, we now include new experiments demonstrating transferability. For transferability we train on a set of 200 dipeptides and infer on 7 dipeptide systems not present in the training set and show accurate free energy estimates and low energy/torus w-2 scores. Visually our energy histograms have nearly perfect overlap and free energy projections have great agreement with the test set. For more details on transferability, refer to our reply to reviewer vcq2.
>
> > The method requires training an additional EBM, whereas other approaches typically train only a single CNF model.
>
> We acknowledge that our method requires training two models instead of one. However, we believe that this is a key strength, rather than a limitation. Flow matching models are good for sampling but hard for likelihood evaluations. EBMs on the other hand provide easy access to likelihoods but are not great for sampling. By combining the two types of model we leverage the best of both worlds.
>
> Moreover, our method design significantly reduces inference-time costs, as evaluating the likelihood of a sample requires only a forward pass through the EBM. This is beneficial as it reduces the constraints on the emulator/neural sampler as it gets rid of the requirement of invertibility for Boltzmann generation. Furthermore, in transferability scenarios, i.e. inferring on systems not present in the training set, the EBM requires only a bit of fine tuning for significant inference time gains. The two model approach is also advantageous when scaling to higher dimensional systems, where training an additional neural network is less computationally expensive than calculating Jacobian trace integrals.
>
> > The architecture choices are complex, yet architectural ablations are missing. A discussion is needed to clarify that performance gains are not merely due to increased parameter counts.
>
> We acknowledge that our flow matching architectures are slightly more complex than the ECNF model (to account for multiple vector features), however, the GVP models our study has significantly fewer parameters (108,933) than the benchmark ECNF (147,599). We now report the parameter counts in our updated paper.
>
> > The endpoint models diverge near t = 0, requiring manual truncation of the integration. This instability reduces the effectiveness of likelihood-based sampling.
>
> We agree with the reviewer that the endpoint model is not well-suited for likelihood-based sampling. However, we do not view this as a limitation of our method. Our approach leverages an EBM to estimate likelihoods, thereby avoiding the computational burden of the continuous change-of-variable equation. A key insight from our method is that introducing a secondary EBM for likelihood estimation relaxes the design constraints on the Emulator models. Since efficient likelihood evaluation is no longer required for Boltzmann generation, the Emulator can be trained using the endpoint objective. We demonstrate this in our experiments on alanine dipeptide and emphasize that this is the central takeaway from our use of the endpoint objective.
>
> > If the Boltzmann emulator is trained on biased data with missing modes, the EBM is unable to recover those missing modes.
>
> We agree with the reviewer regarding the issue of missing modes; however, this is a limitation common to most generative models and not specific to our method. The emulator needs to be able to sample sufficiently from the modes of the dataset for effective reweighting with the EBM. Notably, the authors of the original ECNF paper[1] also reported needing to bias the data toward underrepresented modes in order to obtain accurate free energy differences. Our experimental design follows the same approach.
>
> > Is it possible to just use EBM for sampling, since it already provide a score of the model density.
>
> This is an interesting idea worth exploring further. However, we emphasize that the sample distribution must approximately match the likelihoods predicted by the EBM to achieve the Boltzmann. In practice, achieving this with an EBM typically would require running Langevin dynamics for large amounts of time, which may be computationally infeasible.
>
> > In your dipeptide experiments, how does the model perform if the emulator training dataset are more biased?
>
> We biased our dataset to account for the underrepresented mode along the positive $\varphi$ angle. This step is necessary to obtain accurate free energy differences method. We adopted the same method for biasing as employed in the original ECNF work[1] to ensure a fair and equitable comparison between the two methods. Again, we acknowledge the need for our Emulator model to be able to cover all the modes, but this is a limitation that is common to most generative models and is discussed in previous work as well[1].
>
> [1]Leon Klein, Andreas Krämer, and Frank Noé. Equivariant flow matching. Advances in Neural Information Processing Systems, 36:59886–59910, 2023.
>
> Overall, we thank the reviewer for their thoughtful comments and hope they will consider increasing their score in light of the new experimental results and our responses to their concerns.

---

> > ### Comment · Reviewer_jMkc · 2025-08-06
> >
> > Thank you for your responses. However, I still have some concerns regarding whether the performance improvement is attributable to the method itself rather than the use of GVP. While it is true that there are 30% fewer parameters, this difference could easily be offset by employing more complex networks. As a result, I will maintain my original score.

---

> > > ### Author Response · Authors · 2025-08-06
> > >
> > > Thank you for the comment. We believe there may be a misunderstanding regarding the main objective of our paper. Our goal is to recover the Boltzmann distribution defined by an energy function for a molecular system. A common approach to this task involves using a CNF-based generative model to sample conformers and then reweight them using likelihoods computed via the (continuous) change of variable equation. However, this likelihood calculation becomes prohibitively expensive for large systems.
> > >
> > > To address this, our method replaces the change-of-variable computation with an energy-based model (EBM) that directly predicts likelihoods through a simple forward pass, enabling orders-of-magnitude speedup during inference. This substantial reduction in inference time is also a key contribution of our method.
> > >
> > > | Method             | $\varphi$ $\Delta F$ Error   |Reweighted E-W2    | Reweighted T-W2 | Inference time (h)|
> > > |:--------------------|--------------:|-------------:|-------------------:|-------------------:|
> > > |  ECNF  | 0.03 ± 0.23   |0.37±0.02	  | 0.59±0.00 |9.366|
> > > |  GVP Endpoint | 0.79 ± 2.61   | 2.88±0.01  | 0.58±0.01 | 26.24 |
> > > |  GVP Vector Field | 0.28 ± 0.67   | 0.46±0.05	 |0.60±0.00 | 18.42|
> > > |  BoltzNCE Endpoint | 0.04 ± 0.94   | 2.78±0.04 | 0.59±0.01 |0.164 |
> > > |  BoltzNCE Vector Field | **0.02 ± 0.13**   | **0.27±0.02**	 | **0.57±0.00** |**0.09** |
> > >
> > > We evaluate how well each method captures the Boltzmann distribution using four metrics (also reported in Table 2 of the paper and in our response to reviewer vcq2):
> > > (1) free energy difference error ($\varphi$ $\Delta F$ Error),
> > > (2) reweighted energy-Wasserstein-2 distance (Reweighted E-W2),
> > > (3) reweighted torsion-Wasserstein-2 distance (Reweighted T-W2), and
> > > (4) inference time.
> > >
> > > Notably, the ECNF model actually **outperforms** the GVP-based Boltzmann generators (denoted GVP-\* in the tables) on these metrics. One possible explanation is that the GVP architecture leads to a less accurate ODE integration for the change of variable computation. As such, the increased architectural complexity of GVP models **does not** translate into improved performance.
> > >
> > > In fact, we observe better results only when using the EBM (referred to as BoltzNCE-\*) to predict likelihoods for conformers generated by GVP. Moreover, inference times with BoltzNCE are significantly lower than with ECNF. Together, these results demonstrate that our method is central to the performance improvements over the existing baseline.
> > >
> > > These results demonstrate that our improved performance is not due to the quality of generated samples/choice of network architecture of the sampler, but rather the method of calculating likelihoods with the EBM. We hope that the reviewer appreciates this and considers increasing their score.

---

> > > > ### Comment · Reviewer_jMkc · 2025-08-07
> > > >
> > > > Thank you for addressing my concerns. I will raise my score.

---

### Note · Authors · 2025-08-11

We are grateful for the reviewers' thoughtful feedback, which has helped strengthen our work. Although only three of the four reviewers participated in the discussion, we trust that all found our responses satisfactory. For the camera-ready version, we will integrate the suggested analyses and improvements, including results on transferability, EBM vs. Hutchinson likelihood estimators, E-W2 and T-W2 scores, and coupling functions. We will also refine key sections for clarity, particularly those on the InfoNCE loss function, Umbrella Sampling, the Free Energy Difference metric, and the density chasm problem, and have added targeted citations to related and seminal works.

We believe these updates address the reviewers' concerns and further enhance the paper's clarity, rigor, and impact. We thank the reviewers for their constructive engagement and hope the AC finds the paper a strong candidate for acceptance at NeurIPS 2025.

---

### Decision · Program_Chairs · 2025-09-17

**Decision:**

Accept (poster)

**Comment:**

This paper introduces BoltzNCE, a method that pairs a generative model with an energy-based model to efficiently model Boltzmann distributions for molecular systems. The reviewers collectively recommend accepting the paper which I agree with. The core idea of using an EBM for rapid likelihood estimation is novel and address a significant computational bottleneck. The rebuttal and discussion phase addressed concerns around transferability, strengthening the recommendation for acceptance.